# InduCE: Inductive Counterfactual Explanations for Graph Neural Networks

**Samidha Verma**                                                              *samidha.verma@cse.iitd.ac.in*
*Indian Institute of Technology, Delhi, India*

**Burouj Armgaan**                                                             *burouj.armgaan@cse.iitd.ac.in*
*Indian Institute of Technology, Delhi, India*

**Sourav Medya**                                                                      *medya@uic.edu*
*University of Illinois, Chicago, USA*

**Sayan Ranu**                                                                     *sayanranu@iitd.ac.in*
*Indian Institute of Technology, Delhi, India*

**Reviewed on OpenReview:** *https://openreview.net/forum?id=RZPN8cgqST*

## Abstract

Graph neural networks (GNNs) drive several real-world applications including drug-discovery, recommendation engines, and chip designing. Unfortunately, GNNs are a *black-box* since they do not allow human-intelligible explanations of their predictions. *Counterfactual* reasoning is an effort to overcome this limitation. Specifically, the objective is to minimally perturb the input graph to a GNN, so that its prediction changes. While several algorithms have been proposed towards counterfactual explanations of GNNs, majority suffer from three key limitations: *(1)* they only consider perturbations in the form of deletions of existing edges, *(2)* they perform an inefficient exploration of the combinatorial search space, *(3)* the counterfactual explanation model is *transductive* in nature, i.e., they do not generalize to *unseen* data. In this work, we propose an *inductive* algorithm called INDUCE, that overcomes these limitations. Through extensive experiments on graph datasets, we show that incorporating edge additions, and modelling marginal effect of perturbations aid in generating better counterfactuals among available recourse. Furthermore, inductive modeling enables INDUCE to directly *predict* counterfactual perturbations without requiring instance-specific training. This leads to significant computational speed-up over baselines and allows counterfactual analyses for GNNs at scale.

## 1 Introduction and Related Work

The applications of Graph Neural Networks (GNNs) have percolated beyond the academic community. GNNs have been used for drug discovery Stokes et al. (2020), designing chips Mirhoseini et al. (2020), and recommendation engines Ying et al. (2018). Despite significant success in prediction accuracy, GNNs, like other deep learning based models, lack the ability to explain why a particular prediction was made. Explainability of a prediction model is important towards making it trust-worthy. In addition, it sheds light on potential flaws and generates insights on how to further refine a model.

**Existing Works:** At a high level, GNN explainers can be classified into the two groups of *instance-level* Ying et al. (2019); Luo et al. (2020); Shan et al. (2021); Yuan et al. (2021); Huang et al. (2022); Yuan et al. (2022); Lucic et al. (2022); Tan et al. (2022); Lin et al. (2021); Bajaj et al. (2021); Abrate & Bonchi (2021); Wellawatte et al. (2022) or *model-level* explanations Yuan et al. (2020). Consistent with their nomenclature, instance-level explainers explain a specific input graph, whereas model-level explainers provide a high-level explanation in understanding general behaviour of the GNN model trained over a set of graphs. Recent research has also

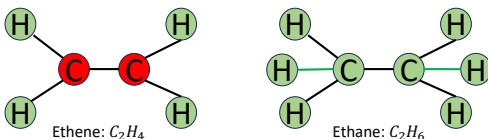

Figure 1: The figure illustrates the graph representations of two molecules. A node is classified as red if it represents an unsaturated carbon atom, meaning the valency of the atom is not saturated by covalent bonds with hydrogen atoms. Otherwise, the node is classified as green. In the counterfactual task of converting an unsaturated carbon atom to a saturated one, we need to add two hydrogen bonds, as indicated by the green edges in Ethane. Counterfactual reasoners that only consider edge deletions would be able to explain saturated carbon atoms with counterfactuals, but not vice versa.

| Method | Explainability Paradigm | Modeling Marginal Impact | Additions | Deletions | Inductive | # explainers required |
|---|---|---|---|---|---|---|
| **GNNExplainer** Ying et al. (2019) | Factual | ✗ | ✗ | ✓ | ✗ | $N$ |
| **PGExplainer** Luo et al. (2020) | Factual | ✗ | ✗ | ✓ | ✓ | 1 |
| **GEM** Lin et al. (2021) | Factual | ✗ | ✗ | ✓ | ✓ | 1 |
| **CF-GNNExplainer** Lucic et al. (2022) | Counterfactual | ✗ | ✗ | ✓ | ✗ | $N$ |
| **CF²** Tan et al. (2022) | Counterfactual + Factual | ✗ | ✗ | ✓ | ✗ | $N$ |
| **RC-explainer** Bajaj et al. (2021) | Counterfactual + Factual | ✗ | ✗ | ✓ | ✓ | 1 |
| **Clear** Ma et al. (2022) | Counterfactual | ✗ | ✓ | ✓ | ✓ | 1 |
| **InduCE (Ours)** | Counterfactual | ✓ | ✓ | ✓ | ✓ | 1 |

Table 1: **Comparison on properties of common perturbation-based GNN explainers. The last column shows the number of required explainers for a graph with $N$ nodes.**

focused on global concept-based Xuanyuan et al. (2023); Azzolin et al. (2023) explainers that provide both model and instance-level explanations. Instance-level methods can broadly be grouped into two categories: *factual* reasoning Ying et al. (2019); Luo et al. (2020); Shan et al. (2021); Yuan et al. (2021); Huang et al. (2022); Yuan et al. (2022) and *counterfactual* reasoning Lucic et al. (2022); Tan et al. (2022); Bajaj et al. (2021); Abrate & Bonchi (2021); Wellawatte et al. (2022). Given the input graph and a Gnn, factual reasoners seek to identify the smallest sub-graph that is sufficient to make the same prediction as on the entire input graph. Counterfactual reasoners, on the other hand, seek to identify the smallest perturbation on the input data that changes the Gnn's prediction. Perturbations correspond to removal and addition of edges.

Compared to factual reasoning, counterfactual reasoners have the additional advantage of providing a means for recourse Voigt & Von dem Bussche (2017). For example, in drug discovery Jiang et al. (2020); Xiong et al. (2021), mutagenicity is an adverse property of a molecule that hampers its potential to become a drug Kazius et al. (2005). While factual explainers can attribute the subgraph causing mutagenecity, counterfactual reasoners can identify this subgraph along with the changes that would make the molecule non-mutagenic. In this work, we study counterfactual reasoning over Gnns. To illustrate our problem, let us consider the input graph shown in Fig. 1. Here, each node belongs to the *green* class if it is part of the motif (subgraph) shown on the right. Otherwise, it belongs to the *yellow* class. The dotted edge on node A does not exist, for now. At this stage, if we ask the counterfactual reasoner to change the label of node A, the best answer would be to add the dotted edge. Similarly, for node B, one possible answer would be to delete the edge marked with ⊗.

Table 1 presents a structured summary of the instance-level counterfactual explainers. In this work, we innovate on the following key directions that lead to improved explanations and efficiency (§ 4).

- **Modeling Marginal Probabilities in the Perturbations Space:** Identifying the minimal set of perturbations that induces a change in graph predictions inherently involves a combinatorial problem. Hence, it is important for the learning methodology to model the marginal influence exerted by individual perturbations on the remaining perturbations. Existing algorithms have largely disregarded this vital aspect. Notably, Cf² Tan et al. (2022), CF-GnnExplainer Lucic et al. (2022) and RCExplainer adopt an approach centered on learning a binary *mask* representing the joint distribution over the feasible perturbations. Clear uses Graph-VAE Simonovsky & Komodakis (2018) to generate a counterfactual graph, thus learning only the joint distribution by optimizing a modified counterfactual loss Ma et al. (2022) over the generated graph. In contrast, InduCE introduces an innovative reinforcement learning-based strategy,

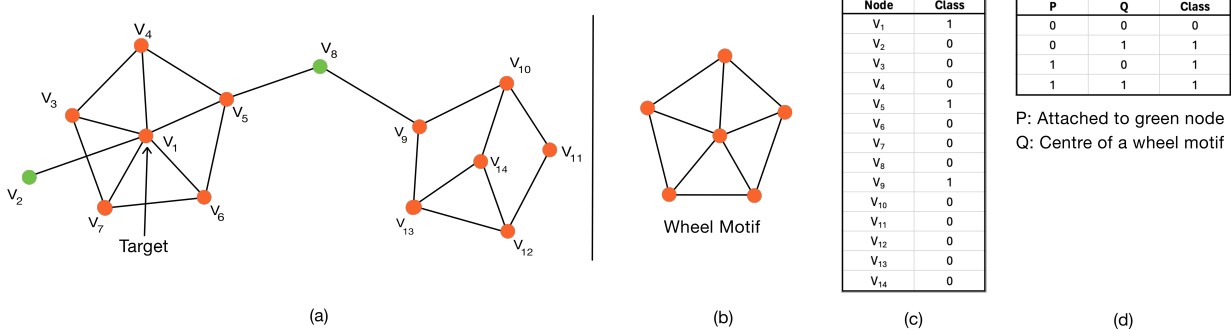

Figure 2: **(a)** A sample graph that highlights the merits of marginal edits over one-shot edits. **(b)** A wheel motif **(c)** Class labels of each node **(d)** Ground truth rules for each class where 0 and 1 mean presence and absence of the property, respectively. The node colors represent the node labels.

wherein we focus on quantifying the marginal effect of an individual perturbation within the context of a given set of other perturbations. This perspective facilitates the exploration of the combinatorial space at a more nuanced level, resulting in demonstrably enhanced outcomes, as substantiated in § 4.

Fig. 2 presents an example showcasing the merits of marginal edits over one shot. Consider the sample graph under binary node classification where node $v_1$, belonging to class 1 is the target node. Assume that in the underlying data, a node belongs to class 1 if it is attached to either a green node or at the center of a wheel motif or both (see the class label function in Fig. 2). Otherwise, a node belongs to class 0. Provided the trained GNN has learned this logic, a counterfactual explanation on $v_1$ would detach the green node from $v_1$ and remove any one of the wheel's spokes (edges between the target and the ring). Now, let us look at the counterfactual generation for the target node under one-shot and marginal settings:

**One-shot:** Explainers that learn a mask for all the edges in parallel will assign an importance value $I_1$ to the edge $\langle v_1, v_2 \rangle$ between the target and the green node, and importance $I_2$ to all the spokes ($\langle v_1, v_3 \rangle$, $\langle v_1, v_4 \rangle$, $\langle v_1, v_5 \rangle$, $\langle v_1, v_6 \rangle$, $\langle v_1, v_7 \rangle$) since all of them have an identical contribution to the target node's predicted class. Since one-shot explainers employ independent selection on each edge for deletion, it either selects all spokes for deletion or none. Thus, when a counter-factual explanation is found, it leads to six edge deletions to get the counterfactual when only two edits should suffice.

**Marginal:** A marginal explainer's first episode will be identical. The edge $\langle v_1, v_2 \rangle$ will have importance $I_1$, and all the spokes will be assigned importance $I_2$. Now, based on these values, the marginal explainer will either pick the $\langle v_1, v_2 \rangle$ edge or one of the spoke edges. Let us consider both cases:

1. $\langle v_1, v_2 \rangle$ is deleted in the first episode. In the next episode, the importance values will be recomputed and all the spokes will, once again, receive the same importance. Now, one of them will be chosen and deleted. At the end of this episode, the explainer will identify that the resultant graph has crossed the decision boundary, and there is no need to proceed and thereby finding an explanation of size 2.

2. One of the spokes with importance $I_2$ is chosen and deleted first. In the next episode, the importance scores will be recomputed for all the edges. The remaining spokes will be assigned identical importance values again, however, it will be a significantly lower value since deleting a spoke no longer has an effect on the GNN's prediction. This is because the marginal explainer will see that removing more spokes has a lower reward than removing the green node's edge. Hence, the explainer will remove the green node's edge. Following this, the explainer will identify that the resultant graph has crossed the decision boundary, and the process will terminate.

In either case, the marginal explainer will return a counterfactual graph with only two edge deletions, which is closer (in this case identical) to the ground truth compared to one-shot's six edits, showing its superior quality.

- **Ability to add edges:** In counterfactual reasoning, the task is to identify the minimal change in graph such that the label changes. This objective is different from factual explanations where the goal is to identify the substructure responsible for a prediction. Indeed, in a factual explainer, edge additions do not have a role since we need to reason within the scope of the structural components within the input

graph being explained. However, in counter-factual reasoning, there is no natural reason to remain constrained to deletions only. Several existing techniques do not consider addition of edges (or nodes); they only consider edge removals. This limitation compromises the search space consisting of possible "changes" on the input graph. Fig. 1 illustrates this limitation with a concrete example. Specifically, an unsaturated hydrocarbon can be converted to a saturated hydrocarbon only through addition of edges.Saturated hydrocarbons are significantly less reactive (and therefore a more stable fuel) than unsaturated hydrocarbons, leading to vastly different properties. Now, consider the counterfactual task of identifying the minimal change in an unsaturated carbon to obtain a saturated hydrocarbon. This task requires adding (saturating) C-H edges to the unsaturated carbon atoms (highlighted in red color). However, existing counterfactual reasoners, with the exception of CLEAR Ma et al. (2022), lack the capability to add edges. We address this gap in our proposed method.

- **Inductive modeling:** Several of the existing counterfactual explainers are transductive in nature (See Table 1), i.e., they cannot generate counterfactuals on unseen nodes. As an example, if the model is trained to generate counterfactuals on node $v$ of graph $G$, it cannot be used to generate counterfactuals on another node $u$ of $G$. Consequently, these transductive models need to be retrained on each node of an input graph. In contrast, an inductive model learns parameters from a train set of nodes, which in turn can be used to *predict* counterfactual on unseen nodes. In addition, an inductive model is robust to changes in the input graph due to external factors such as new friend connections in a social network, citations in a citation network, etc. $\text{CF}^2$ and CF-GNNEXPLAINER cannot be used in an inductive setting for the following reason. These explainers initialize a parameter space based on the number of nodes in the input graph, resulting in an $n \times n$ weight matrix. Subsequently, this matrix is transformed into a binary $n \times n$ mask matrix, which is element-wise multiplied with the original graph's adjacency matrix to denote edge deletion (with 0s in the mask indicating deletion). As a consequence, the model's parameters are contingent upon the graph's node count, making it infeasible to apply on unseen graphs of varying sizes in an inductive setting.

**Contributions:** In this work, we develop INDUCE (Inductive Counter-factual Explanations), that addresses the above limitations of existing counterfactual reasoners. We propose INDUCE to addresses these challenges and make the following contributions:

- **Novel formulation:** We formulate the novel problem of *model-agnostic, inductive* counterfactual reasoning over GNNs for node classification. It is worth noting that both inductive modeling and the ability to add edges introduce non-trivial challenges. In inductive modeling, we need to learn parameters that embodies general rules to be used for predicting counterfactuals. In the transductive approach, since parameters are learned for each specific node, there is no generalization component. Edge additions introduce a significant scalability challenge as the number of possible additions grows quadratically to the number of nodes in the graph. In contrast, the number of edge deletions is $O(|\mathcal{E}|)$, where $\mathcal{E}$ is the set of edges in the graph. (§ 2).

- **Algorithm:** Identifying the smallest number of edge additions or removals that alter the prediction is a combinatorial optimization problem. We prove that computing the optimal solution to the problem is NP-hard (§ 2). As a heuristic, we *learn* to solve this combinatorial optimization problem through reinforcement learning powered by *policy gradients* Williams (1992) (§ 3). Instead of learning the joint distribution over perturbations, the proposed approach sequentially selects the optimal set of perturbations by modeling their marginal effect.

- **Empirical validation:** We show that INDUCE outperforms state-of-the-art algorithms in metrics relevant to counterfactual reasoning. We further analyze the generated counterfactuals and provide compelling evidence that enabling edge additions and modelling the marginal effect of perturbations impart superior performance. Finally, we also showcase the computation gains obtained due to embracing the inductive paradigm instead of transductive modelling (§ 4).

## 2 Problem Formulation

We use the notation $\mathcal{G} = (\mathcal{V}, \mathcal{E})$ to denote a graph with node set $\mathcal{V}$ and edge set $\mathcal{E}$. We assume each node $v_i \in \mathcal{V}$ is characterized by a feature vector $x_i \in \mathbb{R}^d$. Furthermore, $l(v) : v \to \mathcal{C}$ is a function mapping each node $v$ to its true class label drawn from a set $\mathcal{C}$. We assume there exists a GNN $\Phi$ that has been trained on $\mathcal{G}$. Given an input node $v_i \in \mathcal{V}$, we assume $\Phi(\mathcal{G}, v, c)$ outputs a probability distribution over class labels

$c \in \mathcal{C}$. The predicted class label is therefore the class with the highest probability, which we denote as $L_\Phi(\mathcal{G}, v) = \arg\max_{c \in \mathcal{C}} \{\Phi(\mathcal{G}, v, c)\}$.

**Problem 1 (Counterfactual Reasoning on Gnns)** *Given input graph $\mathcal{G} = (\mathcal{V}, \mathcal{E})$, a target node $v \in \mathcal{V}$, a Gnn model $\Phi$, and an optional set of node pairs $\mathcal{V}_c = \{(v_i, v_j) \mid v_i, v_j \in \mathcal{V}\}$ between which edges may be perturbed, find the closest graph $\mathcal{G}^*$ by minimizing the number of perturbations, such that $L_\Phi(\mathcal{G}^*, v) \neq L_\Phi(\mathcal{G}, v)$ and all perturbed edges are among pairs in $\mathcal{V}_c$.*

In a real world, we may not have control over all perturbations. $\mathcal{V}_c$ allows us to specify that. If $\mathcal{V}_c \subseteq \mathcal{E}$, we restrict to only deletions. On the other hand, if $\mathcal{V}_c \cap \mathcal{E} = \emptyset$, we only allow additions.

In our problem, we enforce two restrictions on the counterfactual reasoner. First, it should be *model-agnostic*, i.e., only the output of $\Phi$ is visible to us, but not its parameters. Second, the reasoner should be *inductive*, which means we should learn a *predictive model* $\Pi$, that can predict the counterfactual graph $\mathcal{G}^*$ given the inputs $\mathcal{G}$, Gnn $\Phi$, and target node $v$.

**Theorem 1 (NP-hardness)** *Counterfactual reasoning for Gnns, i.e., Prob. 1, is NP-hard.*

PROOF. To prove NP-hardness of the problem we reduce it from the classical *set cover* problem.

**Definition 1 (Set Cover Feige (1998))** *Given a collection of subsets $\mathcal{S} = \{S_1, \cdots, S_m\}$ from a universe of items $U = \{u_1, \cdots, u_n\}$ identify the smallest collection of subsets $\mathcal{A}^* \subseteq \mathcal{S}$ covering the set $U$, i.e.,*

$$\mathcal{A}^* = \arg\min_{|\mathcal{A}|, \mathcal{A} \subseteq \mathcal{S}} \bigcup_{\forall S_i \in \mathcal{A}} S_i = U \tag{1}$$

We show that given any instance of a set cover problem $\langle \mathcal{S}, U \rangle$, it can be mapped to Prob. 1. Specifically, we construct a graph $\mathcal{G} = (\mathcal{V}, \mathcal{E})$, where $\mathcal{V} = N \cup \mathcal{S} \cup U$. Here, $N$ is an arbitrary set of nodes. In addition, we have a node corresponding to each set $S \in \mathcal{S}$ and each item $u \in U$. There is an edge between two nodes $v_i, v_j \in \mathcal{V}$ if $v_i$ corresponds to some set $S \in \mathcal{S}$, $v_j$ corresponds to item $u \in U$, and $u \in S$. There are no edges among nodes in $N$. The Gnn $\Phi$ predicts the label of any node $v \in N$ as 1 if all nodes from $U$ are reachable from $v$, otherwise 0. Furthermore, let the set of allowed perturbations be $\mathcal{V}_c = \{(v_i, v_j) \mid v_i \in N, v_j \in \mathcal{S}\}$. Given any $v \in N$, with $L_\Phi(\mathcal{G}, v) = 0$, the counterfactual reasoner therefore needs to identify the minimum number of edges to add so that all nodes from $U$ are reachable from $v$ through some nodes in $\mathcal{S}$.

With this construction, only edge additions are allowed. Now it is easy to see that the smallest edge set changing the label of $v$ corresponds to connecting $v$ to nodes in $\mathcal{A}^*$, where $\mathcal{A}^*$ is the solution for the set cover problem. □

Owing to NP-hardness, it is not feasible to identify the closest counterfactual graph in polynomial time. Hence, we aim to design effective heuristics. [1]

# 3    InduCE

Our goal is to learn an inductive counterfactual reasoning model $\Pi$, and thus, the proposed algorithm is broken into two phases: *training* and *inference*. During training, we learn the parameters of the model $\Pi$ and during inference, we predict the counterfactual graph using $\Pi$. Theorem 1 prohibits us from supervised learning since generating training data of ground-truth counterfactuals is NP-hard. Hence, we use reinforcement learning. Through *discounted rewards*, reinforcement learning allows us to model the combinatorial relationships Khalil et al. (2017) in the perturbation space.

## 3.1    Learning $\Pi$ as an MDP

Given graph $\mathcal{G}$, we randomly select a subset of vertices from $\mathcal{V}$ to train $\Pi$. Given a target node $v$, the task of $\Pi$ is to iteratively delete or add edges such that with each perturbation the likelihood of $\Phi(\mathcal{G}^t, v) \neq \Phi(\mathcal{G}, v)$ changes maximally. Here, $\mathcal{G}^t = (\mathcal{V}, \mathcal{E}^t)$ denotes graph $\mathcal{G}$ after $t$ perturbations starting with $\mathcal{G}^0 = \mathcal{G}$. We

---

[1]We note that the proof should not be confused with the NP-hardness proof in Huang et al. (2023), which shows that global counterfactual summarization is NP-hard.

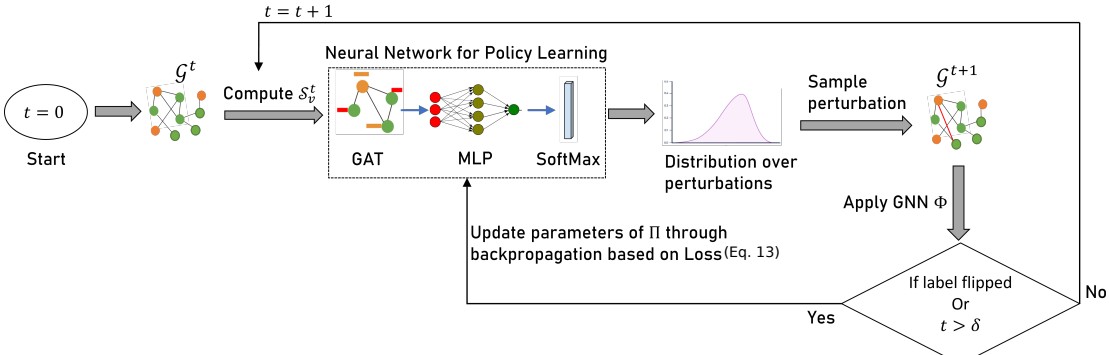

Figure 3: Pipeline of the policy learning in INDUCE. $\delta$ indicates the maximum number of perturbations.

model this task of iterative perturbations as a *markov decision process (MDP)*. Specifically, the *state* captures a latent representation of the graph indicative of how it would react to a perturbation. An *action* corresponds to an edge addition or deletion by $\Pi$. Finally, the *reward* is a function of the number of perturbations, which we want to minimize, and the probability of $\Phi(\mathcal{G}^t, v)$ changing following the next action (edge addition or deletion), a value that we want to maximize. We next formalize each of these notions.

**State:** Intuitively, the state should characterize how likely the class label of the target node $v$ would change following a given action. Towards that end, we observe that a GNN of $\ell$ layers aggregates information from the $\ell$-hop neighborhood of $v$. Nodes outside this neighborhood do not impact the prediction of a GNN. Motivated by this design of GNNs, the state in our problem is the set of node representations in the $h$-hop neighborhood of the target node $v$, where ideally $h \geq \ell$. Specifically, at time $t$, the state is:

$$\mathbf{S}_v^t = \left\{ \mathbf{x}_u^t \mid u \in \mathcal{N}_v^h \right\}, \text{ where} \tag{2}$$

$$\mathcal{N}_v^h = \{ u \in \mathcal{V} \mid sp(v, u) \leq h \} \tag{3}$$

Here, $sp(v, u)$ denotes the length of the shortest path from $v$ to $u$ in the original graph $\mathcal{G} = (\mathcal{V}, \mathcal{E})$.

The representations of nodes, i.e., $\mathbf{x}_u^t$, are constructed using a combination of semantic, topological, and statistical features.

- *Original node Features:* It is common to encounter graphs where nodes are annotated with features or labels (recall the definition of $x_i$ in § 2). We retain these features.
- *Degree Centrality:* The higher the degree of a node, the more information it receives from its neighbors. Thus, when an edge is added or deleted from the target node to a high-degree node, it may have significant impact on the representation of the target. Based on this observation, we use the degree of a node as a part its representation.
- *Entropy:* The entropy of a node at time $t$ is defined as $e_u^t = -\sum_{\forall c \in \mathcal{C}} p_c \log p_c$, where $p_c = \Phi(\mathcal{G}^t, v, c)$. The entropy quantifies the uncertainty of the GNN $\Phi$ on a given node. We hypothesize that if the $\Phi$ is highly certain (i.e., low entropy) about the class label of some node $u$, then any perturbation on $u$ is unlikely to make it change. Similarly, the opposite is true on nodes with high entropy. Due to this information content of entropy, we use it as one of the features in $\mathbf{x}_u^t$.
- *Class label:* Finally, we include the predicted class label of a node, i.e., $L_\Phi(\mathcal{G}^t, u)$ in the form of one-hot encodings of dimension $\mathcal{C}$.

The final representation of node $u$ and time $t$ is therefore the concatenation of the above features, i.e.,

$$\mathbf{x}_u^t = x_i \parallel degree_u^t \parallel e_u^t \parallel \left( \text{one-hot} \left( L_\Phi \left( \mathcal{G}^t, u \right) \right) \right) \tag{4}$$

Here, $\parallel$ represents the *concatenation* operator.

**Actions:** The action space consists of all possible edge deletions in the $h$-hop neighborhood of target node $v$ and additions of edges from $v$ to other non-attached nodes in its $h$-hop neighborhood. Formally, the sets

are defined as follows:

$$\mathcal{E}_{v,del}^t = \left\{ e = (u_i, u_j) \in \mathcal{E}^t \mid u_i, u_j \in \mathcal{N}_v^h \right\} \tag{5}$$

$$\mathcal{E}_{v,add}^t = \left\{ e = (v, u_j) \notin \mathcal{E}^t \mid u_j \in \mathcal{N}_v^h \right\} \tag{6}$$

The action space is the perturbation set:

$$\mathcal{P}^t = \mathcal{E}_{v,del}^t \cup \mathcal{E}_{v,add}^t \tag{7}$$

**Reward:** Our objective is to change the predicted label of target node $v$ with the minimum number of perturbations in $\mathcal{N}_v^h$ in order to find the counterfactual. To capture these intricacies, we formulate the reward of an action $a$ as a combination of the prediction accuracy of GNN $\Phi$ and the number of perturbations made so far.

$$\mathcal{R}_v^t(a) = \frac{1}{\mathcal{L}_{v,pred}^{t+1} + \beta \times d(\mathcal{G}, \mathcal{G}^t)}, \text{ where} \tag{8}$$

$$\mathcal{L}_{v,pred}^t = \sum_{\forall c \in \mathcal{C}} \mathbb{1}_{l(v)=c} \log \left( \Phi \left( \mathcal{G}^t, v, c \right) \right), \qquad d(\mathcal{G}, \mathcal{G}^t) = t + 1 \tag{9}$$

In simple terms, $\mathcal{L}_{v,pred}^{t+1}$ is the *log-likelihood* of the data predicted by $\Phi$ in $\mathcal{G}^{t+1}$ on $v$. $\mathcal{G}^{t+1}$ is the created upon perturbing $\mathcal{G}^t$ with action $a$. $\beta$ is a hyper-parameter that regulates how much weight is given to log-likelihood of the data vs. the perturbation count. $d$ is the distance function, which in our case is simply the number of edge edits made to $\mathcal{G}$ at time step $t$.

**State Transitions:** At time $t$, the action corresponds to selecting a perturbation $a \in \mathcal{P}^t$ (Recall Eq. 7) from $p_{a,v}^t \sim \Pi \left( a \mid \mathcal{S}_v^t \right)$. We will discuss the computation of $p_{a,v}^t$ in § 3.2.

### 3.2 Neural Architecture for Policy Training

To learn $p_{a,v}^t$, we we take the representations in $\mathbf{S}_v^t$, and pass them through a neural network comprising of a $K$-layered *Graph Attention Network* (GAT) (Veličković et al., 2018), an MLP, and a final softmax layer. The GAT learns a $d$-dimensional representation $\mathbf{a} \in \mathbb{R}^d$ for each perturbation $a \in \mathcal{P}_v^t$. $\mathbf{a}$ is then passed through an *Multi-layered Perceptron (MLP)* to embed them into a scalar representing their value, which is finally passed over a softmax layer to learn a distribution over $\mathcal{P}_v^t$. The entire network is trained end-to-end. We next detail each of these components.

GAT: Let $\forall u \in \mathcal{N}_v^h, \mathbf{h}_u^0 = \mathbf{x}_u^t$ (Recall Eq.4). In each layer $k \in [1, K]$, we perform the following transformation:

$$\mathbf{h}_u^k = \sigma \left( \sum_{\forall u' \in \mathcal{N}_u^1 \cup \{u\}} \alpha_{u,u'}^k \mathbf{W}^k \mathbf{h}_{u'}^{k-1} \right) \tag{10}$$

$\sigma$ is an activation function, $\alpha_{u,u'}^k$ are learnable, layer-specific attention weights, and $\mathbf{W}^k \in \mathbb{R}^{d^{k-1} \times d^k}$ is a learnable, layer-specific weight matrix where $d^k$ is the hyper-parameter denoting the representation dimension in hidden layer $k$. In our implementation, we use LeakyReLU with negative slope 0.01 as the activation function. The attention weights are learned through an MLP followed by a softmax layer. Specifically,

$$e_{u,u'}^k = \texttt{MLP}(\mathbf{h}_u^{k-1} \parallel \mathbf{h}_{u'}^{k-1}), \text{ where } e_{u,u'}^k \in \mathbb{R},$$

$$\alpha_{u,u'}^k = \frac{exp \left( e_{u,u'}^k \right)}{\sum_{\hat{u} \in \mathcal{N}_u^1 \cup \{u\}} exp \left( e_{u,\hat{u}}^k \right)}$$

After $K$ layers, the GAT outputs the final representation $\mathcal{X}_u = \mathbf{h}_u^K$ for each node $u$ in $v$'s neighborhood. Semantically, given the initial state representation $\mathbf{x}_u^t$, the GAT enriches them further by merging with topological information. Finally, the representation of an action $a \in \mathcal{P}_v^t$ is set to $\mathbf{a} = \mathcal{X}_u \parallel \mathcal{X}_v \parallel t(u, v)$, where:

$$t(u, v) = \begin{cases} 0, & a \in \mathcal{E}_{v,del}^t \text{ (Recall Eq. 5)} \\ 1, & a \in \mathcal{E}_{v,add}^t \text{ (Recall Eq. 6)} \end{cases} \tag{11}$$

**MLP and SoftMax layers:** The value of $a$ is $s_a = \mathtt{MLP}(\mathbf{a})$, where $s_a \in \mathbb{R}$. Finally, we get a distribution over all actions in $\mathcal{P}_v^t$ as:

$$p_{a,v}^t = \Pi(a \mid \mathcal{S}_v^t) = \frac{exp(s_a)}{\sum_{\forall a' \in \mathcal{P}_v^t} exp(s_{a'})} \tag{12}$$

### 3.3 Policy Loss Computation

We iteratively sample an action as per Eq. 12 till either the label changes or we exceed the maximum number of perturbations (which is a hyper-parameter). This iterative selection generates a trajectory of perturbations $\mathcal{T}_v = \{a_1, \cdots, a_m\}$. We use the standard loss for policy gradients on $\mathcal{T}_v$ Williams (1992). More specifically, we minimize the following loss function:

$$\mathcal{J}(\Pi) = -\frac{1}{\mathcal{V}_{tr}} \left( \sum_{\forall v \in \mathcal{V}_{tr}} \left( \sum_{t=0}^{|\mathcal{T}_v|} \log p_{a,v}^t \mathcal{R}_v^t(a_t) + \eta Ent(\mathcal{P}_v^t) \right) \right) \tag{13}$$

Here, $\mathcal{V}_{tr} \subseteq \mathcal{V}$ is the subset of nodes on which the RL policy is being trained. $Ent(\mathcal{P}_v^t)$ is the entropy of the current probability distribution over the action space.

$$Ent(\mathcal{P}_v^t) = -\sum_{\forall a \in \mathcal{P}_v^t} p_{a,v}^t \log(p_{a,v}^t) \tag{14}$$

By adding the entropy to the loss, we encourage the RL agent to explore when there is high uncertainty. $\eta$ is a hyper-parameter balancing the *explore-exploit* trade-off. For simplicity of exposition, we omit the discussion on discounted rewards in Eq. 13. Discounted rewards better capture the combinatorial relationship in the perturbation space. Refer to App. B for details.

### 3.4 Training and Inference

Fig. 3 presents the training pipeline. Starting from the original graph, we compute the state representation at each iteration $t$. The state is passed to the neural network to compute a distribution over the perturbation space. A perturbation is sampled from this distribution and the graph is accordingly modified. The GNN $\phi$ is then applied on the modified graph. If the label changes or the number of perturbation exceeds the maximum limit, we update the policy parameters. Otherwise, we update the state and continue building the perturbation trajectory in the same manner. The pseudocode of the training pipeline is provided in Alg.2 (Refer to App. A).

**Inductive inference:** In INDUCE, each edit involves either adding or deleting an edge. The embedding of such an edge (or action) is formed by concatenating the embeddings of its connected nodes. These node embeddings are learned through a GAT operating on the current graph state. Since GAT is inherently inductive—meaning their parameter set is invariant to changes in graph size — Induce inherits this property. This is demonstrated in Eq. 10, where the parameter set dimension is bounded by $\mathbf{W}^k \in \mathbb{R}^{d^{k-1} \times d^k}$. Here, $d^k$ signifies the hidden dimension in layer $k$ and is independent of the graph size. In essence, the embedding of an action (such as edge addition or deletion) is influenced by the local $\ell$-hop neighborhood surrounding the two nodes involved in the edit. Consequently, even when presented with an unseen graph, as long as the training data contains similar distributions of neighborhood topologies, the predictive accuracy of INDUCE remains robust.

Alg. 1 presents the pseudocode of the inference pipeline. We iteratively make forward passes till the label changes or we exceed the budget. The forward pass is identical to the training phase with the only exception being we deterministically choose the perturbation with the highest likelihood instead of sampling.

**Transductive Inference:** This phase proceeds identical to the training phase with the only exception that we learn a target node specific policy instead of one that generalizes across all nodes.

### 3.5 Complexity of InduCE

**Train-Phase:** Training the policy network involves four key steps, i.e., compute the state, forward pass through the policy network, sample and take action, and compute the marginal reward of the action. Among

---

**Algorithm 1** Test pipeline of INDUCE.

---

**Input:** Graph $\mathcal{G}$, GNN $\Phi$, target node $v$, maximum perturbation budget $\delta$
**Output:** Counterfactual explanations CF

1: CF $= \phi$
2: $\Pi \leftarrow$ parameters of pre-trained policy
3: $t \leftarrow 0$
4: **while** $L_\Phi(\mathcal{G}^0, v) = L_\Phi(\mathcal{G}^t, v)$ & $t < \delta$ **do**
5:      compute $\mathcal{S}_v^t$
6:      $a^* \leftarrow \arg\max_{a \in \mathcal{P}^t} \Pi(\mathcal{S}_v^t, \mathcal{P}^t)$
7:      $\mathcal{G}_v^{t+1} \leftarrow$ perturb $\mathcal{G}_v^t$ with edge $a^*$
8:      $t \leftarrow t + 1$
9: **if** $L_\Phi(\mathcal{G}^0, v) \neq L_\Phi(\mathcal{G}^t, v)$ **then**
10:      CF $=$ CF $\cup \{a^*\}$
11: **Return** CF

---

the above, state computation and performing the action take $\mathcal{O}(1)$ time. Time taken by a forward pass through the policy network involves a combination of computing node embeddings using GAT and computing a score for each action in the action space $\mathcal{P}^t$ through the MLP. Forward pass through the GAT takes $\mathcal{O}(K(|\mathcal{V}|h_i h_d + |\mathcal{E}|h_d))$ (Veličković et al., 2018) where $K$, $h_i$ and $h_d$ are the number of GAT layers, input and hidden dimensions respectively. Action score computation using MLP takes $\mathcal{O}(|\mathcal{P}^t|h_m(h_d + (J - 2)h_m + 1)))$, $J$ and $h_m$ are number of MLP layers and hidden dimensions. Computing the reward function involves taking a forward pass through a GCN which takes $\mathcal{O}(L|\mathcal{E}|h_i h_d|\mathcal{C}|)$ time, here $L$ and $|\mathcal{C}|$ are the number of layers and number of classes respectively. The above steps are repeated for $M$ episodes for each node in training set $V_{tr}$ until a maximum of $\delta$ time steps. Combining the above costs, treating $J$, $K$, $L$, $h_i$, $h_d$, $h_m$ and $|\mathcal{C}|$ as fixed constants which have small values, and with the knowledge that $|\mathcal{P}^t| = \mathcal{O}(|\mathcal{V}| + |\mathcal{E}|)$ (refer Eq. 7), the complexity with respect to the input parameters reduces to $\mathcal{O}(|\mathcal{V}_{tr}|M\delta(|\mathcal{V}| + |\mathcal{E}|))$.

**Test-Phase:** The test phase involves state computation, forward pass through the policy network, and performing the action with highest probability for a maximum of $\delta$ time-steps for every node in the test set $\mathcal{V}_{test}$. Therefore following the discussion in the training phase, the time complexity of test phase is $\mathcal{O}(|\mathcal{V}|_{test}|\delta(|\mathcal{V}| + |\mathcal{E}|))$.

## 4 Experiments

In this section, we benchmark INDUCE against established baselines. The anonymized code base and datasets used in our evaluation are submitted with supplementary material. The hardware and software platform details are provided in App. D.

### 4.1 Datasets

| | Tree-Cycles | Tree-Grid | BA-Shapes | Amazon | ogbn-arxiv |
|---|---|---|---|---|---|
| # Classes | 2 | 2 | 4 | 6 | 40 |
| # Nodes | 871 | 1231 | 700 | 397 | 169,343 |
| # Edges | 1950 | 3410 | 4100 | 2700 | 1,166,243 |
| Motif size (# nodes) | 6 | 9 | 5 | NA | NA |
| Motif size (# edges) | 6 | 12 | 6 | NA | NA |
| # Nodes from motifs | 360 | 720 | 400 | NA | NA |
| Avg node degree | 2.23 | 2.77 | 5.86 | 15.90 | 6.89 |

Table 2: The statistics of the benchmark datasets.

**Benchmark Datasets:** We use the same three benchmark graph datasets used in Tan et al. (2022); Lin et al. (2021); Lucic et al. (2022). Statistics of these datasets are listed in Table 2. Each dataset has an undirected base graph with pre-defined motifs attached to random nodes of the base graph, and randomly added additional edges to the overall graph. The class label of a node indicates whether it is part of a node or not. Further details on the datasets are provided in App. D.1.

**Real Dataset:** We additionally use real-world datasets from the Amazon-photos co-purchase network Shchur et al. (2018) and ogbn-arxiv Wang et al. (2020). In the Amazon dataset, each node corresponds

| Method | Tree-Cycles | | | Tree-Grid | | | BA-Shapes | | |
|---|---|---|---|---|---|---|---|---|---|
| | Fid.(%) ↓ | Size ↓ | Acc.(%) ↑ | Fid.(%) ↓ | Size ↓ | Acc.(%) ↑ | Fid.(%) ↓ | Size ↓ | Acc.(%)↑ |
| Random | **0** | 3.18 ±2.32 | 67.08 | **0** | 8.32 ±4.95 | 73.44 | **0** | 283.97 ±272.76 | 15.57 |
| CF-GnnEx | 49.0 | 1.05 ±0.23 | **100** | 10 | 1.37 ±0.58 | 92.24 | 37.0 | 1.31 ±0.55 | **95.83** |
| Cf² | 76.38 | 4.18 ±1.89 | 67.68 | 98.45 | 5.5 ±1.5 | 44.64 | 23.68 | 4.10 ±1.64 | 70.54 |
| InduCE (trans.) | **0** | **1.01 ±0.12** | 98.61 | **0** | **1.02 ±0.12** | **97.67** | **0** | **1.30 ±0.90** | 95.31 |
| CF-GnnEx ++ | 100 | NULL | NULL | 100 | NULL | NULL | 38.16 | 6315.44 ±9916.50 | 17.36 |
| Cf² ++ | 13.89 | 28.34 ±7.56 | 19.24 | 28.68 | 12.90 ±7.71 | 27.44 | 100 | NULL | NULL |
| InduCE (trans.)−− | **0** | 1.40 ±1.49 | 81.94 | **0** | 1.24 ±0.43 | 92.64 | 6.6 | 1.42 ±1.49 | 83.22 |

Table 3: **Results for transductive methods:** Lower fidelity, smaller size, and higher accuracy are desired. "NULL" denotes that the method could not produce a counterfactual. The best results are in bold. Tr., Fid., Acc. denote transductive, fidelity and accuracy, respectively. The best result in each category is highlighted in bold. The second best in each category is underlined.

to a product, edges correspond to products that are frequently co-purchased, node features encode bag-of-words from product reviews and the node class label indicates the product category. The ogbn-arxiv dataset is a citation network. The nodes are all computer science arXiv papers indexed by MAG Wang et al. (2020). Each directed edge represents that one paper cites another. The features are word embeddings of the title and the abstract computed by the skip-gram model Mikolov et al. (2013). The labels are subject areas. Since the class labels in these datasets are not based on presence or absence of motifs, the corresponding cells in Table 2 are marked as "NA".

## 4.2 Baselines

We benchmark INDUCE against the state-of-the-art baselines of **(1)** CF-GNNEXPLAINER Lucic et al. (2022), **(2)** CF² Tan et al. (2022), **(3)** RCEXPLAINER Bajaj et al. (2021) and **(4)** CLEAR Ma et al. (2022). In addition, we also compare against the state-of-the-art factual explainers **(5)** PGEXPLAINER Luo et al. (2020) and **(6)** GEM (Lin et al., 2021) to show that when factual explainers are used for counter-factual reasoning by removing the factual explanation (subgraph) from the input graph, they are not effective. This is consistent with prior reported literature Lucic et al. (2022); Tan et al. (2022); Lin et al. (2021). Finally, we also compare against **(5)** RANDOM perturbations. While CF² and CF-GNNEXPLAINER are transductive, RCEXPLAINER, CLEAR, GEM and PGEXPLAINER are inductive. The codebase of all algorithms, except RCEXPLAINER (Refer App. J) have been obtained from the respective authors. We provide the hyper-parameter settings for all the baselines in the App D.2. A detailed analysis of how the hyper-parameters affect the performance of INDUCE is provided in App. K.

## 4.3 Performance Measures

To quantify performance, we use the standard measures from the literature Lucic et al. (2022).

- **Fidelity:** Fidelity is the percentage of nodes whose labels do not change when the edges produced by the explainer (algorithm) are perturbed. Lower fidelity is better. Furthermore, it may be argued that fidelity is the most important metric among the three measures.
- **Size:** Explanation size is the number of edges perturbed for a given node. Lower size is better.
- **Accuracy:** Accuracy is the percentage of explanations that are correct. As standard in CF²,CF-GNNEXPLAINER, and GEM, this translates to the percentage of edges in the counterfactual that belong to the motif. Since nodes have a non-zero class label only if they belong to a motif, the explanation for nodes should be edges in the motif itself. Note that accuracy is computable only on the benchmark datasets since they include ground-truth explanations.
- **Sparsity:** Sparsity is inversely related to size. We present the definition and results in App. E due to space constraints.

**Other settings:** Details of experimental settings, the black-box GNN, training and inference are in App. D.3.

| Method | Tree-Cycles | | | Tree-Grid | | | BA-Shapes | | |
|---|---|---|---|---|---|---|---|---|---|
| | Fid.(%) ↓ | Size ↓ | Acc.(%) ↑ | Fid.(%) ↓ | Size ↓ | Acc.(%) ↑ | Fid.(%) ↓ | Size ↓ | Acc.(%)↑ |
| PGExplainer | 34.72 | 6 | 76.85 | 41.09 | 6 | 66.93 | 6.58 | 6 | 89.25 |
| Gem | 95 | 6 | 88.97 | 97 | 6 | 94.57 | 17 | 6 | **98.44** |
| RCExplainer | 98.61 | **1.0 ±0.0** | 70.73 | 100 | NULL | NULL | NA | NA | NA |
| Clear | 56.94 | 47.92 ±21.84 | 86.17 | **0** | 184.16 ±34.29 | **96.34** | DNS | DNS | DNS |
| InduCE (ind.) | **0** | 2.31 ±1.44 | **96.57** | **0** | **4.67 ±2.91** | 91.05 | **2.6** | 4.37 ±3.53 | 64.40 |
| InduCE (ind.) – – | 36.3 | 1.67 +- 0.90 | 90.32 | 16.3 | 6.38 ±3.74 | 86.31 | 40.8 | **3.37 ±3.04** | 56.08 |

Table 4: **Results for inductive methods.** The best result in each category is highlighted in bold. The second best in each category is underlined. "DNS" denotes that the method did not scale "NA" means not applicable as RCExplainer code did not extend to multi-class setting.

## 4.4 Quantitative Results

**Transductive methods:** Table 3 presents the results (for now, we will focus on the first four rows). Our method InduCE in the transductive setting outperforms all the baselines almost in all settings. For Tree-Cycles and BA-Shapes, CF-GnnExplainer is producing better accuracy. However, we note that its fidelity is much worse, indicating it fails to find an explanation more frequently. While CF-GnnExplainer consistently achieves the lowest size among the baselines, its fidelity is much worse. This indicates that CF-GnnExplainer is able to solve only the easy cases and hence the low size is deceptive as it did not solve the difficult ones.

**Inductive methods:** Table 4 shows that InduCE is superior to Gem and PGExplainer in most cases. The fidelity scores produced by Gem and PGExplainer are much higher (worse). This indicates, in most of the cases, Gem and PGExplainer are unable to find a counterfactual example. Also the explanation size is fixed in Gem and PGExplainer since they work with fixed budgets. Clear performs worst in terms of average explanation size. This defeats the purpose of counterfactual explanations since the perturbations should be minimal so that the instance just crosses the decision boundary. This is necessary to make the explanations human-intelligible as well. It has non-zero fidelity as it significantly perturbs the target nodes' local neighbourhoods, hence leading to a change in the predicted labels. The high accuracy is possibly due to a large number of edge additions but is meaningless in the face of a significantly large counterfactual size. Moreover, it does not scale for the dense datasets, BA-Shapes. Conclusively, generative modelling used by Clear, leads to complex explanations owing to large edits in the graph structure. Thus, modelling the marginal impact of perturbations through policy gradients is an effective approach. RCExplainer did not extend to multi-class setting. It has high fidelity, low size and accuracy, indicating that it only finds counterfactuals for easy cases.

**Transductive vs Inductive:** We further compare the inductive version (Table 4) of our method, InduCE with the transductive baselines (Table 3). While the transductive methods have a clear advantage of re-training the model instance wise, the results produced by InduCE-inductive are comparable. As noted earlier, although CF-GnnExplainer achieves better size than InduCE-inductive, its fidelity is much worse indicating that the low size is a manifestation of not being able to explain the hard cases that InduCE is able to explain. Moreover, in addition to the ability to generalize to unseen nodes, inductive modeling also imparts a dramatic speed-up in generating explanations (see Table 6a).

**Impact of motif topology:** The complexity of the motif topology, along with the inherent difficulty of the prediction task, significantly influences the prediction accuracy of the Gnn being explained and hence also affects the explainer. Specifically, when the Gnn's predictions are poor due to the complexity of the task, there may be inherent uncertainty in the model functionality. This uncertainty can pose challenges for the explainer, as it must contend with ambiguity and variability in the model being explained when proposing counterfactual explanations. The explainer may struggle to provide confident and reliable explanations in the face of such uncertainty, leading to potentially inconsistent or unreliable results.

Furthermore, the larger the size of the motif, the more is the expected size of the explanation to make a non-motif node part of a motif. This trend is visible in Table 4, where the explanation size of Induce is largest for Tree-grid dataset, where the motif consists of 9 edges.

Finally, the shape also plays a role. The expressivity of a message-passing GNN is bounded by 1-WL tests Xu et al. (2019). If the GNN cannot distinguish between two shapes, this ambiguity will be passed on to the explainer as well.

**Additional experiments:** App. F contains experiments on the impact of **(1)** heuristic features and **(2)** the choice of GNN architectures. Experiments on counterfactual size vs accuracy trade-off are in App. H.

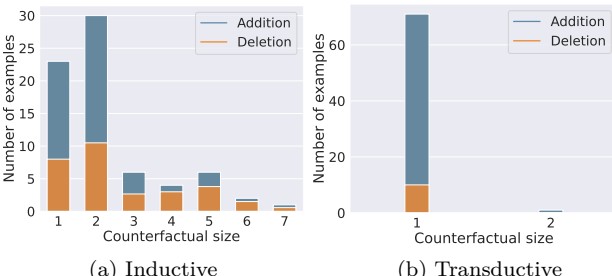

|  (a) Inductive | (b) Transductive |

Figure 4: The distributions of the edit size and their internal composition of edge additions and deletions by INDUCE on the Tree-Cycles dataset.

## 4.5 Impact of Edge Additions

We seek answers to two key questions: **(1)** How much does the performance of INDUCE deteriorate if we restrict edge additions? **(2)** If we empower the baselines also with additions, do they match up to INDUCE? To answer the first question, we study the performance of INDUCE in the setting where only edge deletions are allowed. The rows corresponding to INDUCE (transductive)−− and INDUCE (inductive)−− in Tables 3 and 4 present these results. It is evident that the deletion-only version produces inferior results for both the transductive and inductive versions. In Fig. 4, we further study the frequency distribution of edge additions and deletions in the counter-factual explanations produced by INDUCE in Tree-Cycles dataset (results on other datasets are in App. G). We observe that additions dominate the perturbations, and thereby, further establishing its importance, which INDUCE unleashes. To address the second question, we empower CF-GNNEXPLAINER and $CF^2$ with edge additions, denoted as CF-GNNEXPLAINER ++ and $CF^2$ ++ respectively. [2] Both $CF^2$ and CF-GNNEXPLAINER use a *mask-based* strategy. A mask is a learnable binary matrix of the same dimension as the $\ell$-hop neighborhood of the target node. By taking an element-wise product of the mask with the adjacency matrix, one obtains the edges to be deleted. When empowered with additions, the mask itself becomes the new adjacency matrix. Surprisingly, the performance of CF-GNNEXPLAINER drops, while for $CF^2$, we see improvement in fidelity in two out of three datasets. Further investigation into this performance reveals that edge additions significantly increase the search space of possible perturbations (See Table I in Appendix). A mask-based strategy is a single-shot learnable paradigm that does not examine the marginal effect of each perturbation. When the perturbation space increases, it overwhelms the learning procedure. In contrast, INDUCE uses reinforcement learning where a trajectory of perturbations is selected based on their marginal gains. This allows better modeling of the combinatorial nature of counterfactual reasoning. Overall, the above experiments reveal that both additions, as well as an algorithm equipped to model large combinatorial spaces, are required to perform well.

## 4.6 Impact of Modeling Marginal Impact

INDUCE adopts an auto-regressive framework where the selection of edits depends on previously selected edits. This design is adopted since, as established in Theorem 1, counterfactual reasoning is a combinatorial problem. *What happens if we ignore modeling the marginal impact of edits?* We investigate this aspect

---

[2] GEM is not extendible to additions (See App. D.2 for details), PGEXPLAINER does not incorporate perturbations with the intent of changing the label since it is a factual explainer.

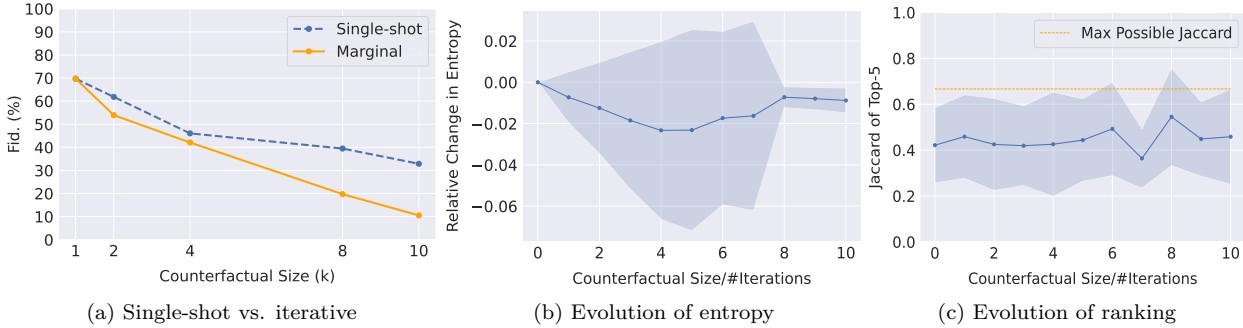

(a) Single-shot vs. iterative          (b) Evolution of entropy          (c) Evolution of ranking

Figure 5: (a) Fidelity comparison of INDUCE with single-shot prediction on the BA-Shapes dataset. In single-shot prediction, we select the $k$ edits with highest predicted reward directly. Recall that lower fidelity is better. (b) Evolution of the entropy of the score distribution over perturbation space against the number of iterations (explanation size). (c) Evolution of the top-5 highest scoring edits against the number of iterations (explanation size).

in Fig. 5a. As visible, INDUCE achieves significantly lower fidelity than its *single-shot* counterpart, thus emphasizing the importance of modeling the marginal impact of edits. In single-shot prediction, we directly add the top-$k$ edits with the highest expected reward based on the original graph state. This contrasts with the auto-regressive pipeline where a single edit with the highest expected reward is first added to the original graph, and then all available edits are re-prioritized based on their expected rewards on the edited graph. To ensure a fair comparison, $k$ in the single-shot approach is set to the number of edits made by INDUCE until the node label changes.

We next analyze the evolution of the entropy of the perturbation space in Fig. 5b. We note a decrease in entropy indicating consolidation of perturbation preference till a certain number of iterations, beyond which, it starts to increase again. This trend is not entirely surprising, since a higher number of iterations indicates that INDUCE has failed to change the label of the target node despite a significant number of edits, and such cases correspond to higher amount of uncertainty

*How does the ranking of the edits based on their expected rewards change over iterations?* Fig. 5c analyzes this question. Specifically, we compute the top-$k$ highest ranked edits at each timestep $t$, starting from $t = 0$, and compute its Jaccard similarity to the top-$k$ list at $t + 1$. Note that since we select the highest ranked edit in each timestep, and therefore this edit cannot be present in the next timestep, the maximum Jaccard similarity between the lists at two consecutive time steps is $(k - 1)/(k + 1)$. In Fig. 5c, we note that the Jaccard similarity is consistently lower than this upper bound, indicating that the ranking list is evolving over time. $k$ is set to 5 in this analysis. Connecting back to Fig. 5a, this further highlights the need for modelling perturbations marginally instead of single-shot.

## 4.7 Quantitative Results on Real Datasets

| Method | Amazon | | ogbn-arxiv | |
|---|---|---|---|---|
| | **Fid.(%)** ↓ | **Size** ↓ | **Fid.(%)** ↓ | **Size** ↓ |
| **CF-GnnEx** | 100 | NULL | DNS | DNS |
| **Cf$^2$** | 60 | 13.7 +- 16.98 | DNS | DNS |
| **InduCE (trans.)** | **53.50** | **4.72 $\pm$ 4.38** | **0** | **1.00 $\pm$ 0.00** |

Table 5: Results for transductive methods on the real-world dataset. "NULL" denotes that the method could not produce a counterfactual. "DNS" denotes that the method could not produce a counterfactual as it did not scale. Refer to App. I that details reasons on why these baselines failed to scale.

In Tables 5a and 5b, we present the results of INDUCE and other baselines on the real-world datasets. Consistent with the performance on benchmark datasets, INDUCE continues to outperform all the baselines almost in both transductive and inductive settings. We note that most of the baselines failed to produce

| Method | Amazon | | ogbn-arxiv | |
|---|---|---|---|---|
| | Fid.(%) ↓ | Size ↓ | Fid.(%) ↓ | Size ↓ |
| **PGExplainer** | 100 | NULL | 95.50 | 4 |
| **Gem** | 100 | NULL | DNS | DNS |
| **Clear** | DNS | DNS | DNS | DNS |
| **InduCE (ind.)** | **93.00** | **6.60 ± 2.87** | **78.7** | **3.11** ±3.04 |

Table 6: Results for inductive methods on the real-world dataset. "NULL" denotes that the method could not produce a counterfactual. "DNS" denotes that the method could not produce a counterfactual as it did not scale. Refer to App. I that details reasons on why these baselines failed to scale.

| Method | Tree-Cycles | Tree-Grid | BA-Shapes |
|---|---|---|---|
| **PGExplainer** | 0.41 | 0.62 | 0.38 |
| **Gem** | 0.16 | 0.73 | 8.64 |
| **CF-GnnEx** | 1295.66 | 2382.51 | 3964.36 |
| **$Cf^2(\alpha = 0.6)$** | 304.13 | 154.88 | 2627.09 |
| **RCExplainer** | 0.76 | 1.42 | NA |
| **CLEAR** | 0.06 | 0.10 | DNS |
| **InduCE (ind.)** | 4.36 | 17.64 | 68.33 |
| **InduCE (trans.)** | 66.08 | 331.58 | 6546.48 |

(a) Efficiency

| Dataset | #Nodes | #Edges | Avg. degree | Time/node (ms) |
|---|---|---|---|---|
| **Tree-Cycles** | 871 | 1,950 | 2.23 | 60.56 |
| **Tree-Grid** | 1,231 | 3,410 | 2.77 | 13.67 |
| **BA-Shapes** | 700 | 4,100 | 5.86 | 89.91 |
| **ogbn-arxiv** | 169,343 | 1,166,243 | 6.89 | 353.43 |
| **Amazon-photos** | 7,487 | 119,043 | 15.90 | 5242.32 |

(b) Scalability

Table 7: (a) Running times (in seconds) of each algorithm on entire test set. (b) Scalability against various graph properties. "DNS" denotes that the method did not scale. RCExplainer code does not extend to multi-class setting, we denote this as "NA".

counterfactuals in Amazon, while CLEAR fails to scale. In ogbn-arxiv, on the other hand, all baselines except PGExplainer fails to scale; they crash with out-of-memory exception. In contrast, INDUCE produces promising performance with the transductive version achieving 0% fidelity.

## 4.8 Efficiency

Table 6a presents the inference times of various algorithms. First, the inductive methods (INDUCE, CLEAR, RCEXPLAINER, PGEXPLAINER and GEM) are much faster than the others. Between the inductive methods, PGEXPLAINER and CLEAR are the fastest. INDUCE-inductive is slower since the search space for INDUCE is larger due to accounting for both edge additions and deletions. CLEAR is significantly faster than INDUCE-inductive because it generates a perturbed adjacency matrix for the input graph rather than modelling the marginal effect of edge perturbations. It should be noted that CLEAR performs significantly worse in quantitative metrics and does not scale for dense and large datasets (Recall Table 4). Second, INDUCE-inductive is up to 79 times faster than the transductive methods such as CF-GNNEXPLAINER and $Cf^2$. This speed-up is a result of only doing forward passes through the neural policy network, whereas, transductive methods learn the model parameters on each node separately. Even the transductive version of INDUCE is faster than the other transductive methods for Tree-Cycles and Tree-Grid.

**Scalability against graph size:** Table 6b presents the inference time per node across all datasets. We observe that INDUCE scales to million-sized networks such as ogbn-arxiv. We observe that the growth of the running time is closely correlated with the neighbourhood density, i.e., the average degree of the graph, and not the graph size. In a GNN with $\ell$ layers, only the $\ell$-hop neighbourhood of the target node matters.

| Method | Mutagenicity | | Mutag | | AIDS | |
|---|---|---|---|---|---|---|
| | **Fid.(%)** ↓ | **Size** ↓ | **Fid.(%)** ↓ | **Size** ↓ | **Fid.(%)** ↓ | **Size** ↓ |
| **Cf²** | 38.7 | 2.87 ±1.36 | 88.2 | **1.25 ±0.43** | 98.5 | 4.83 ±2.19 |
| **InduCE (trans.)** | **20.2** | 1.30 ±0.81 | **0** | 1.35 ±0.64 | **67.8** | 4.61 ±2.38 |
| **RCExplainer** | 39.2 | **1.02 ±0.25** | 47.1 | 2.72 ±3.4 | 91.2 | **1.0 ±0.0** |
| **Clear** | OOM | OOM | 64.7 | 16.08 ±6.3 | 86.2 | 296.36 ±36.57 |
| **InduCE (ind.)** | 45.1 | 5.32 ±3.85 | 20.60 | 5.0 ±3.42 | 91.20 | 8.91 ±3.91 |

Table 8: Results on graph classification. The best result in each category is highlighted in bold. The second best in each category is underlined.

## 4.9 Case Study: Counter-factual Visualization

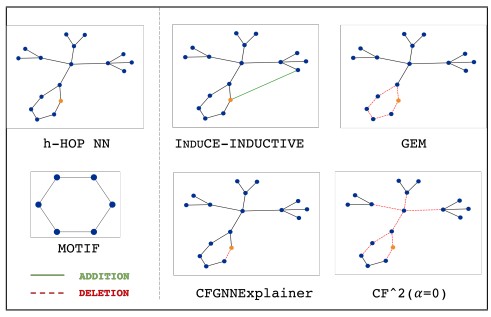

(a) Tree-Cycles

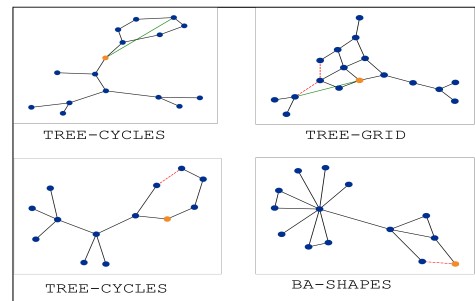

(b) Impact of additions

Figure 6: Visualization of counterfactual explanations for the same node (orange) produced by different methods. Semantically, the node label should change if it is not a part of the motif. **(a)** Counterfactual explanation for Tree-Cycles Dataset, **(b)** Counterfactuals predicted by INDUCE.

In this section, we visually showcase how counterfactual explanations reveal GNN vulnerabilities and the importance of edge additions.

**Revealing GNN vulnerabilities:** A sample counterfactual explanation by various algorithms on Tree-Cycles dataset is provided in Fig. 6a. The target node is part of a motif (6-cycle) and therefore the expected counter-factual explanation is to make it a non-member of a 6-cycle. CF-GNNEXPLAINER correctly finds on such explanation by deleting an edge. Both GEM and Cf² recommend a much larger explanation than necessary. In contrast, INDUCE adds an edge. More interestingly, the target node continues to remain part of the motif. This uncovers a limitation of GNN since it falsely classifies the target node as a non-motif node although it is not. Furthermore, this limitation is uncovered only since INDUCE can add edges. Similar observations in other datasets are available in Figs. Ga-Gb in Appendix.

**Impact of additions:** In Fig. 6b top-left, we share an example where INDUCE changes the label of a target node (orange) by making it part of a cycle through edge additions. Since baselines are only capable of deletes, they fail to change the label of such nodes. However, INDUCE highlights that the GNN does not learn cycle-motifs of specific lengths. In the top-right (Tree-Grid), INDUCE breaks the grid motif and connects the target node (orange) to a non-motif neighbour, hence colluding its embeddings and changing its label. These examples show the importance of edge addition to intuitively explain how the black-box GNN works.

## 4.10 Performance on Graph Classification

While our focus is on explaining node classification, INDUCE can be generalized to graph classification as well. To adapt INDUCE pipeline for explaining graph classification, we use the entire graph to construct the MDP state instead of the $\ell$-hop neighborhood of the target node. Moreover, the action space is defined by all pairs of nodes in the graph since there is no notion of a target node. The rest of the pipeline remains unchanged.

Table 8 presents the results on graph classification explainability on the Mutagenicity Riesen & Bunke (2008); Kazius et al. (2005), Mutag Ivanov et al. (2019) and AIDS Ivanov et al. (2019) datasets. As visible Induce-transductive achieves the best fidelity, while being marginally higher in explanation size, when compared to the smallest size achieved by any of the baselines.

## 5 Concluding Insights

Despite being a black-box, Graph neural networks (GNNs) are becoming the go-to tool for topological data analysis due to their superlative prediction accuracy. The ability to explain predictions is critical towards making a model trustworthy. In this work, we proposed INDUCE to understand GNNs via counterfactual reasoning for the node classification task. While several algorithms in the literature produce counterfactual explanations of GNNs, they suffer from restricted or inefficient counterfactual space exploration and transductivity. INDUCE provides a boost to counterfactual analysis on GNNs by unleashing the power of edge additions, efficient exploration of the combinatorial search space through marginal rewards and inductively predicting explanations on unseen nodes. The proposed features not only lead to better explanations but also provide a significant speed-up allowing INDUCE to perform counterfactual analysis at scale.

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

| Dataset | k-hop | #additions | #deletions | ratio (#additions/#deletions) |
|---|---|---|---|---|
| **Tree-cycles** | 4 | 18313 | 1105 | 16.57 |
| **Tree-grid** | 4 | 98942 | 3960 | 24.98 |
| **BA-shapes** | 4 | 3140886 | 47580 | 66.01 |
| **Amazon-Photos** | 4 | 12144800 | 432160 | 28.10 |
| **ogbg-arxiv** | 3 | 354496362 | 524330 | 676.09 |

Table I: Ratio of no. of additions to deletions of datasets.

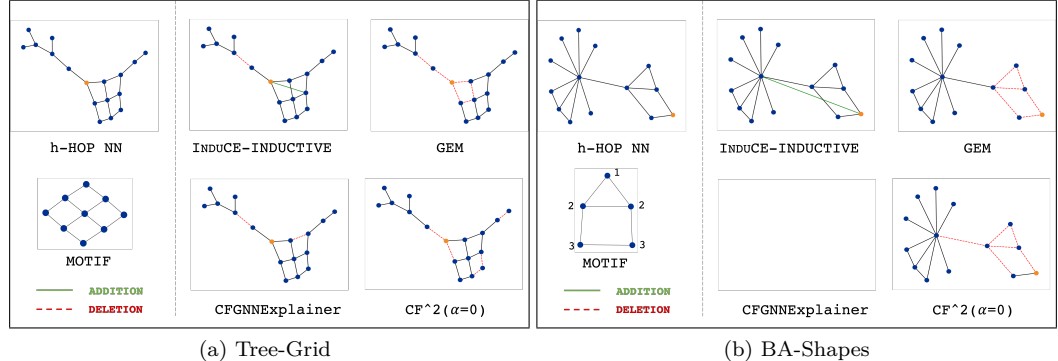

(a) Tree-Grid        (b) BA-Shapes

Figure G: Visualization of counterfactual explanations for the same node (orange) produced by different methods. Semantically, the node label should change if it is not a part of the motif, i.e., **(a)** $3 \times 3$-grid, and the **(b)** the house respectively. CF-GNNEXPLAINER is unable to find a counterfactual in **(b)**.

# A  Appendix

## A  InduCE Training

The training pipeline of INDUCE is given in Algorithm 2.

## B  Discounted Rewards

The objective of the policy $\Pi$ is to find minimal counterfactual explanations for GNNs using the reward function mentioned in Eq 10. However, we can observe that in that equation, the marginal reward at each step is given equal weight. In our case, we want the immediate rewards to have higher weight over the rewards encountered later on in the perturbation trajectory $\mathcal{T}_v$ in order to penalize larger counterfactual size, thus we use discounted rewards (Eq 15) with $\gamma$ being the discount factor to achieve the objective. Since we want minimal explanation size, we use small values of $\gamma$ (Refer App. D.3).

$$\mathcal{R}_{dis,v}^{t}(a_t) = \sum_{i=0}^{\delta} \gamma^i \mathcal{R}_v^{t+i+1}(a_t) \tag{15}$$

here $\delta$ is the maximum perturbation budget.

One limitation of policy gradient is high variance caused by the scale of rewards. A common way to reduce variance is to subtract a baseline, $b(S_v^t)$ such that it does not induce bias in the policy gradient. A simple baseline can be the mean of the discounted rewards, so that we train the policy to pick trajectories that give rewards better than the average rewards. We also normalize the discounted reward further by dividing with the standard deviation.

$$\tilde{\mathcal{R}}_{dis,v}^{t}(a_t) = \frac{\mathcal{R}_{dis,v}^{t}(a_t) - \bar{\mathcal{R}}_{dis,v}^{t}(a_t)}{\max(\sigma(\mathcal{R}_{dis,v}^{t}(a_t)),\ c)} \tag{16}$$

---

**Algorithm 2** Training pipeline of INDUCE.

---

**Input:** Graph $\mathcal{G}$, GNN $\Phi$, Train set $\mathcal{V}_{tr}$, perturbation budget $\delta$, number of episodes $M$
**Output:** Policy $\Pi$

1: $\mathcal{V}_{batch} \leftarrow \{\mathcal{V}_1, \mathcal{V}_2 \ldots, \mathcal{V}_B | \cup_{i=1}^{B} \mathcal{V}_i = \mathcal{V}_{tr}\}$          ▷ Random partitioning instances of $\mathcal{V}_{tr}$ into $B$ batches
2: $\Pi \leftarrow$ initialize with random parameters
3: **for all** $e \in [1, M]$ **do**
4:     **for all** $\mathcal{V}_b \in \mathcal{V}_{batch}$ **do**
5:         **for all** $v \in \mathcal{V}_b$ **do**
6:             $t \leftarrow 0$
7:             **while** $L_\Phi(\mathcal{G}^0, v) = L_\Phi(\mathcal{G}^t, v) \ \& \ t < \delta$ **do**
8:                 compute $\mathcal{S}_v^t$
9:                 $a^t \leftarrow$ sample from $\Pi(\mathcal{S}_v^t, \mathcal{P}^t)$
10:                 $\mathcal{G}_v^{t+1} \leftarrow$ perturb $\mathcal{G}_v^t$ with edge $a^t$
11:                 $\mathcal{R}_v^t \leftarrow$ compute reward using Eq.10
12:                 $\mathcal{R}_{dis,v}^t \leftarrow$ compute discounted rewards using Eq.15
13:                 $\tilde{\mathcal{R}}_{dis,v}^t \leftarrow$ normalize discounted rewards using Eq.16
14:                 $t \leftarrow t + 1$
15:         Backpropagate to minimize loss using Eq. 17
16: **Return** $\Pi$

---

| Dataset | Train accuracy(%) | Test Accuracy(%) |
|---|---|---|
| **Tree-cycles** | 91.23 | 90.86 |
| **Tree-grid** | 84.34 | 87.44 |
| **BA-shapes** | 96.61 | 98.57 |
| **Amazon-Photos** | 89.59 | 88.75 |
| **ogbn-arxiv** | 71.59 | 55.07 |
| **Mutagenicity** | 85.37 | 85.38 |
| **Mutag** | 90.84 | 87.18 |
| **AIDS** | 99.57 | 99.25 |

Table J: Accuracy of GNN $\Phi$ on datasets used in this study.

where $\sigma(\mathcal{R}_{dis}^t)$ is the standard deviation, $c$ is a constant. The optimized version of the loss function(Eq 13) is in App. C.

## C  Batching

Equation 13 takes an average gradient over all examples in the training set. This setting may lead to over-smoothing of the gradients and hence induce difficulty in training. To counter this issue, we performed batching of node instances, and back-propagated with the average gradients computed on nodes in the batch. Thus we optimize our policy on batches of nodes, and use normalized discounted rewards in the loss function (Refer Eq. 17).

$$\mathcal{J}(\Pi) = -\frac{1}{\mathcal{V}_{batch}} \left( \sum_{\forall v \in \mathcal{V}_{batch}} \left( \sum_{t=0}^{|\mathcal{T}_v|} \log p_{a,v}^t \tilde{\mathcal{R}}_{dis,v}^t(a_t) + \eta Ent(\mathcal{P}_v^t) \right) \right) \tag{17}$$

## D  Experimental Setup

All reported experiments are conducted on an NVIDIA DGX Station with four V100 GPU cards having 128GB GPU memory, 256GB RAM, and a 20 core Intel Xeon E5-2698 v4 2.2 Ghz CPU running in Ubuntu 18.04.

### D.1 Benchmark datasets

- **BA-SHAPES:** The base graph is a Barabasi-Albert (BA) graph. The motifs are **house-shaped** structures made up of 5 nodes (Refer Figure Gb). Non-motif nodes are assigned class 0, while nodes at the top, middle, and bottom of the motif are assigned classes 1, 2, and 3, respectively.
- **TREE-CYCLES:** The base graph is a binary tree with **6-node cycles** used as motifs (Refer Figure 6a). The motifs are connected to random nodes in the tree. Non-motif nodes are labelled 0, while the motif nodes are labelled 1.
- **TREE-GRID:** The base graph is a binary tree and the motif is a $3 \times 3$ **grid** connected to random tree nodes (Refer Figure Ga). Just like tree-cycles dataset binary class labelling has been done.

### D.2 Baselines

- **CF-GnnExplainer** Lucic et al. (2022): Being a transductive method for counterfactual explanations, it learns a new set of parameters for every node and cannot be used to explain unseen nodes. We use the default hyperparameters used by the authors.
- **Cf$^2$** Tan et al. (2022): While being transductive in nature, it combines both counterfactual and factual properties to give an explanation. CF$^2$ tunes the parameter $\alpha$ to weigh the contribution of factual explanations. We compare CF$^2$ with $\alpha = 0$ where it becomes as a counterfactual explainer.
- **Gem** Lin et al. (2021): This is inductive by nature, however, it only considers edge deletions. It has a limitation that it learns a factual explanation model where the number of perturbations is fixed, i.e., it does not minimize the number of perturbations with the sole focus on changing the label. We use the default size of 6 as the perturbation size as recommended by the authors.
  Note that GEM is not extendable to include edge additions. Specifically, GEM has a distillation process that generates the ground truth. Distillation involves removing every edge in a node's neighbourhood iteratively and seeing its effect on the loss. The deletions are then sorted based on their effect on the loss. The top-$k$ edges ($k$ is user-specified) are used as the distilled ground truth. The explainer is later trained to generate graphs that are the same as the distilled ground truth. To extend this process for additions, the number of possible edge edits is significantly higher and the iterative process of GEM to create the distilled ground truth does not scale. In addition, it is also unclear how to set $k$ in the presence of additions.
- **PGExplainer** Luo et al. (2020): This method is also inductive and only considers edge deletions. It is a factual explainability method and requires a fixed explanation size as a hyper-parameter. We use the default size of 6 as the perturbation size as recommended by the authors for the benchmark datasets. We also use size 6 and 4 for Amazon-Photos and ogbn-arxiv, respectively.
- **RCExplainer** Bajaj et al. (2021): This method learns explanations that are stable when the input graph is minimally perturbed. The method is inductive and combines, both counterfactual and factual properties. The method has a hyper-parameter $\lambda$ which is used to weigh the contribution of the factual explanations. We set $\lambda = 0$ for comparison of methods in the counterfactual setting.
- **Clear** Ma et al. (2022): CLEAR is an inductive, generative counterfactual explainer. It is capable of node addition or deletion, node feature perturbation and edge additions as well as deletions. The authors' code prioritizes graph classification explainability. We adapt it for node classification by using $k$-hop ego networks for target nodes and training the explanater on these networks. We use the authors' default hyperparameters.
  CLEAR, along with all other baselines do not model the marginal impact of perturbations, which empirically produces better explanations (Refer Table 4, 6), due to the nuanced exploration of the combinatorial search space.
- **Random:** We use the same baseline as used in Lucic et al. (2022). It makes the choices of deleting an edge randomly by generating a random subgraph mask for the $h$-hop neighbourhood of the node and perturbing it.

### D.3 Training and Parameters

**Counter-factual task:** We provide a node that is part of a motif to the counterfactual explainer, and the the task is to change its label by recommending changes in the graph. All nodes that are part of a motif,

are given a specific label and non-motif nodes are given a different label. Since the nodes are always chosen from motifs, the explanation is the motif itself. This setup is identical to $\textsc{Cf}^2$ and CF-GNNEXPLAINER.

**The GNN model $\Phi$:** We use the same GNN model used in CF-GNNEXPLAINER and $\textsc{Cf}^2$. Specifically, it is a Graph Convolutional Networks trained on each of the datasets. Each model has 3 graph convolutional layers, with 20, 128 and 256 hidden dimensions for the benchmarking datasets, Amazon-photos and ogbn-arxiv respectively. For graph classification we use a GNN with 3 lconvolutional layers and 20 hidden dimensions. To compute the graph embedding we do a max-polling over the node embeddings. The non-linearity used is *relu* for the first two layers and *log* softmax after the last layer of GCN. The learning rate is 0.01. The train and test data are divided in the ratio 80:20 for benchmark datasets. For ogbn-arxiv, we use the standard splits provided in the ogb package. In our experiments, we use a scaled-down version of the Amazon-Photos dataset. We choose one random node as the central node and took its $3-$hop neighbourhood in our dataset. Amazon Photos has an average degree of 13, hence, the $3-$hop neighborhood covers a reasonable distribution of class labels. We split the nodes of this subgraph in the ratio of $80 : 20$ for train and test sets. The accuracy of the GNN model $\Phi$ for each dataset is mentioned in Table J.

**Training, Inference and Parameters:** For INDUCE and GEM, we use a train/evaluation split of 80/20 on the benchmark, graph and the Amazon-Photos datasets. For ogbn-arxiv, we train on 10 random examples per class, and sample 1000 random nodes as the test dataset. We make sure that the test and train sets are disjoint. The evaluation set for all techniques are identical. For GEM and INDUCE, the train set is identical. Since CF-GNNEXPLAINER and $\textsc{Cf}^2$ are transductive, only the evaluation set is used for them where they learn a node-specific parameter set. The same happens on the transductive version of INDUCE.

**Parameters settings:** We use $h = 4$ because extracting the 4-hop neighbourhood as the subgraph ensured that we preserve the black-box model's accuracy. We use $\beta = 0.5$ so as to give equal weight to the predict loss and distance loss (see Eq. 10). We use different values of $\gamma \in \{0.4, 0.6\}$ and find the best performance at $\gamma = 0.4$ for the inductive setting and $\gamma = 0.6$ for the transductive setting with a maximum perturbation budget $\delta = 15$. We use maximum number of episodes $\mathbb{M} = 80, 500, 500$ for BA-shapes, Tree-cycles and Tree-grid respectively. We use GAT as the GNN of choice for the policy network. For the policy network, we use 3 GAT layers, 2 fully connected MLP layers, 16 hidden dimension, a learning rate of .0003 and LeakyReLU with negative slope 0.1 as the activation function. We use three different values for $\eta \in \{0.1, 0.01, 0.001\}$ and $\eta = 0.1$ improves the performance due to higher weight for exploration.

| Method | Sparsity (Tree-Cycles) | Sparsity (Tree-Grid) | Sparsity (BA-Shapes) | Amazon-photos | ogbn-arxiv |
|---|---|---|---|---|---|
| **CF-GnnEx** | **0.93** | 0.95 | 0.99 | NULL | DNS |
| **$\textsc{Cf}^2$** | 0.52 | 0.59 | 0.92 | 0.99 | DNS |
| **InduCE (trans.)** | 0.92 | **0.96** | **0.98** | **0.99** | **0.78** |

Table K: **Comparison of "sparsity" of counterfactuals predicted by transductive methods. "NULL" means the baseline could not find a counterfactual. "DNS" means that the baseline did not scale.**

| Method | Sparsity (Tree-Cycles) | Sparsity (Tree-Grid) | Sparsity (BA-Shapes) | Amazon-photos | ogbn-arxiv |
|---|---|---|---|---|---|
| **PGExplainer** | 0.34 | 0.64 | 0.61 | NULL | **0.66** |
| **Gem** | 0.54 | 0.77 | 0.88 | NULL | DNS |
| **RCExplainer** | 0.97 | 0.98 | NA | NA | NA |
| **Clear** | 0.71 | 0.77 | DNS | DNS | DNS |
| **InduCE (ind.)** | **0.90** | **0.88** | **0.99** | **0.99** | 0.64 |

Table L: **Comparison of "sparsity" of counterfactuals predicted by inductive methods. "NULL" means the baseline could not find a counterfactual. "DNS" means that the baseline did not scale (please refer to App. I for detailed reasoning). "NA" means not applicable as RCExplainer code did not extend to multi-class setting (Refer App. J for details.)**

## E  Additional Results on Sparsity of Counterfactuals

Sparsity is defined as the proportion of edges in $\mathcal{N}_v^l$, i.e., the $\ell$-hop neighbourhood of the target node $v$. Since counterfactuals are supposed to be minimal, a value close to 1 is desired. We compare INDUCE

with its baselines on sparsity in Tables K and L. We observe that INDUCE produces better or comparable explanations in terms of sparsity. In Table L we observe that the sparsity of INDUCE-inductive is slightly less than PGEXPLAINER for the ogbn-arxiv dataset. However, the fidelity of PGEXPLAINER is greater than INDUCE-inductive (Recall Table 6). Thus, when looking at the combined results, one may conclude that PGEXPLAINER finds counterfactual explanations for the easier examples and, as a result, has sparser explanations. Similarly, we can interpret the sparsity of CF-GNNEXPLAINER being better than INDUCE-transductive for Tree-Cycles in Table K as its fidelity is much higher than the latter (Recall Table 3).]

## F Ablation Study

To infuse more information about the local graph structure and and its statistics, we use several heuristic features such as degree, entropy, and one-hot encoded labels (Refer 3.1). We conduct an ablation study to investigate the effectiveness of each heuristic feature. Table N summarises the findings. Our method is most consistent when it uses all features. Note that *features* and *entropy* together produce competitive results. However, the fidelity in BA-Shapes becomes much worse from this combination. This means, in most of the cases, this combination is unable to find the counterfactual example. In such cases, the possibility of getting better values in other measures increases.

**Gcn Vs. Gat:** INDUCE uses a GAT to train the RL policy. In the next experiment, we evaluate the impact of replacing the GAT with a Graph Convolutional Network (GCN). The results are presented in Table M. We see that GAT significantly outperforms GCN in the inductive version and thereby justifying our choice.

**Heuristic Features:** Table N contains an exhaustive analyses of the performance of INDUCE-INDUCTIVE using all combinations of heuristic features mentioned in section 3. The combination of features and entropy seems to allow best performance of the model on Tree-grid and Tree-cycles datasets, however as we see in Table 2 that BA-shapes is a dense network and clearly the degree heuristic in combination with node features leads to excellent performance for BA-shapes in terms of the size and the accuracy. Since the ability of the method to find a counterfactual weighs more, our default model containing all heuristic combined with node features gives best overall performance in fidelity, with size and accuracy being better or comparable to the other combinations in most cases.

## G Additional Results on Counterfactual Size Distributions

In figures H and I we observe the distributions of edit distance between the original and the counterfactual $h$-hop neighbourhood of instances in Tree-Grid and BA-Shapes datasets respectively. As observed both in inductive and transductive versions of INDUCE most of the counterfactuals are of small size and dominated by edge additions. However, we can also observe that the transductive versions of INDUCE does produce counterfactuals of size mostly localised around 1. This is because the parameters are tailored instance by instance. INDUCE-inductive however with a minor trade-off in the counterfactual size, provides a comparable performance to INDUCE-transductive (recall Tables 3 and 4) while providing a speed-up of 79x over all the transductive baselines (recall Table 7a). We further conduct experiments on how the accuracy of the explainer is affected with increasing counterfactual size in App. H.

| Policy Variant | Tree-Cycles | | |
| --- | --- | --- | --- |
| | Fid.(%) ↓ | Size ↓ | Acc.(%) ↑ |
| **InduCE-inductive-Gcn** | **0** | 2.62 ±1.52 | 78.84 |
| **InduCE-inductive-Gat** | **0** | **1.99 ±1.00** | **97.47** |
| **InduCE-transductive-Gcn** | **0** | **1.08 ±0.27** | **96.84** |
| **InduCE-transductive-Gat** | **0** | **1.08 ±0.27** | 96.20 |

Table M: Importance of attention in the GNN component of INDUCE.

| Method | Tree-Cycles | | | Tree-Grid | | | BA-Shapes | | |
|---|---|---|---|---|---|---|---|---|---|
| | Fid.(%) ↓ | Size ↓ | Acc.(%) ↑ | Fid.(%) ↓ | Size ↓ | Acc.(%) ↑ | Fid.(%) ↓ | Size ↓ | Acc.(%) ↑ |
| Features only | **0** | 3.12 ±1.96 | 80.05 | **0** | 4.50 ±3.16 | 73.12 | 18.4 | 3.62 ±3.46 | 70.29 |
| Features + D | **0** | 2.56 ±1.73 | 65.74 | **0** | 3.71 ±2.51 | 84.26 | 9.2 | 4.02 ±4.27 | **98.89** |
| Features + E | **0** | **2.24 ±1.15** | 72.69 | **0** | 3.37 ±2.22 | **94.63** | 68.4 | **1.25 ±0.83** | 86.25 |
| Features + OH | **0** | 3.19 ±1.83 | 79.70 | 2.3 | 4.10 ±3.13 | 87.17 | 28.9 | 4.63 ±3.41 | 32.75 |
| Features + D + E | **0** | 2.81 ±1.55 | 85.19 | **0** | **3.12 ±1.96** | 84.77 | 48.7 | 2.43 ±2.65 | 63.71 |
| Features + D + OH | **0** | 2.65 ±1.58 | 69.31 | **0** | 3.16 ±2.21 | 92.02 | 1.3 | 3.62 ±2.45 | 62.08 |
| Features + E + OH | **0** | 2.62 ±1.52 | 78.84 | **0** | 3.47 ±2.28 | 89.15 | 27.6 | 4.2 ±3.02 | 56.00 |
| Features + D + E + OH | **0** | 2.31 ±1.44 | **96.65** | **0** | 4.67 ±2.91 | 91.05 | **2.6** | 4.37 ±3.53 | 64.4 |

Table N: Ablation study results. D, E, and OH represents degree, entropy, and one hot encoded labels respectively. We vary node features along with different heuristic features to measure the effect of each of these features. Our proposed method INDUCE is superior when it uses all the features.

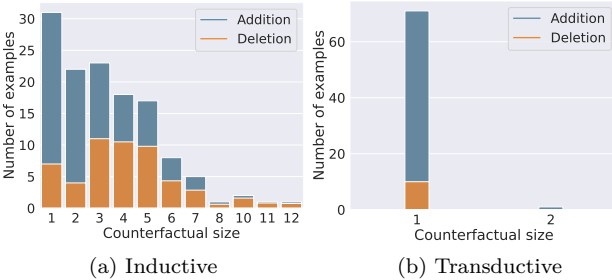

(a) Inductive      (b) Transductive

Figure H: The distributions of the edit size and their internal composition of edge additions and deletions by INDUCE on the Tree-Grid dataset.

## H Size vs. Accuracy Trade-off

The accuracy vs. counterfactual size trade-off for InduCE in Table O and P. We observe that with higher size, the accuracy decreases. Recall, we use benchmark datasets with ground-truth explanations where a node belongs to a particular class if it belongs to a certain motif. Hence, an explanation is accurate if it includes edges from the motif. We observe that when the explainer fails to find short explanations, it typically deviates towards a sequence of edges outside the motif. Hence the explainer fails to change the label till a large set of edits are made.

| Size | Acc. % (Tree-Cycles) | Acc. % (Tree-Grid) | Acc.% (BA-Shapes) |
|---|---|---|---|
| 1 | 100 | 100 | 100 |
| 3 | 94 | 97 | 100 |
| 5 | 73 | 89 | 100 |
| 6 | 83 | NA | NA |
| 7 | 86 | 88 | 100 |
| 10 | NA | 80 | 100 |
| 15 | NA | 73 | 73 |

Table O: **Counterfactual Size vs. Accuracy Trade-off for InduCE- inductive:** The results suggest that as counterfactual size increases, the accuracy of the explanation decreases. NA stands for counterfactual of that size was not present.

| Size | Acc. % (Tree-Cycles) | Acc. % (Tree-Grid) | Acc. % (BA-Shapes) |
|---|---|---|---|
| 1 | 97 | 98 | 100 |
| 2 | 100 | 100 | 100 |
| 6 | NA | NA | 50 |
| 7 | NA | NA | 57 |

Table P: **Counterfactual Size vs. Accuracy Trade-off for InduCE- transductive:** The results suggest that as counterfactual size increases, the accuracy of the explanation decreases. NA stands for counterfactual of that size was not present.

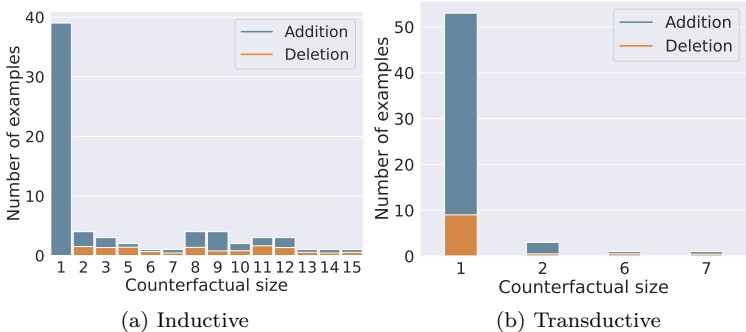

Figure I: The distributions of the edit size and their internal composition of edge additions and deletions by INDUCE on the BA-Shapes dataset.

## I   Non-scalability of the Baselines

**Most baselines do not scale for Ogbn-arxiv dataset.   We describe the details as follows.** Ogbn-arxiv is a **million-sized** node prediction dataset with **169,343** nodes and **1,166,243** edges. $CF^2$, GEM and CLEAR do not scale (Recall Tables 5a and 5b) on this dataset since they employ computations on a dense adjacency matrix, which require $\mathcal{O}(n^2)$ space, where $n$ is the number of nodes in the graph. For a million-sized graph, this leads to memory overflow. Adapting to a sparse adjacency matrix requires non-trivial changes to the source code.

CF-GNNEXPLAINER extracts the $k-$hop neighbourhood of a target node at the runtime and adapts to sparse adjacency matrices more easily.  CF-GNNEXPLAINER's algorithm is model-agnostic, however, the code-base is suitable for its customized black-box and cannot be trivially extended to any other black-box. The explainer loads the black-box weights into itself before freezing them, assuming that the black-box uses the same architecture as itself. It then uses those weights rather than the black-box GNN during the explanation.  In case the explainer's architecture does not match the black-box architecture, the keys for loading the weights do not match, hence, the explainer's weights are not loaded, rather randomly initialized. As a result, the explainer gets initialized with random weights rather than the black-box's weights and acts as a random classifier.  This is an inefficient design choice and prevents CF-GNNEXPLAINER's code from scaling for ogbn-arxiv(Recall Table  5). We use PyTorch-geometric's Fey & Lenssen (2019) standard *GCNConv* layers Kipf & Welling (2016) that are compatible with sparse adjacency matrices to scale the black-box GNN to the million-sized graph.  INDUCE's code is written in a model-agnostic fashion.  Any black-box GNN which employs *log* softmax non-linearity at the last layer is compatible with INDUCE.

**Clear further does not scale for dense networks BA-Shapes and Amazon** (Recall Table  4 and 5b. The reason is that for dense graphs $k$-hop neighbourhoods are significantly larger. The backbone of CLEAR is graph-VAE Simonovsky & Komodakis (2018) which does not scale for large graphs.

## J   Results of RCExplainer on Benchmark datasets

The codebase provided by the authors of RCEXPLAINER could be utilized for explanation of graph classification only. We modified it for node classification by taking the $k$-hop ego-networks of nodes and training the explainer with these networks, however, the results were not reproducible. Hence, we implemented the code for the binary node classification task. The explainer failed to push the node across the decision boundary to get a counterfactual and was stuck in an infinite loop. We further limited the loop to a finite number of iterations. The results are reported on the benchmark binary classification datasets in Table  4. We observe the predicted counterfactuals are sub-par to INDUCE-inductive.

| $\beta$ | Tree-Cycles | | | Tree-Grid | | | BA-Shapes | | |
|---|---|---|---|---|---|---|---|---|---|
| | Fid.(%) ↓ | Size ↓ | Acc.(%) ↑ | Fid.(%) ↓ | Size ↓ | Acc.(%) ↑ | Fid.(%) ↓ | Size ↓ | Acc.(%)↑ |
| 0.0001 | 0 | 1.01 ±0.12 | 97.22 | 0 | 1.02 ±0.12 | 96.12 | 0 | 1.21 ±0.67 | 97.87 |
| 0.25 | 0 | 1.01 ±0.12 | 97.22 | 0 | 1.02 ±0.12 | 96.90 | 0 | 1.30 ±0.86 | 94.34 |
| 0.5 (default) | 0 | 1.01 ±0.12 | 98.61 | 0 | 1.02 ±0.12 | 97.67 | 0 | 1.30 ±0.90 | 95.31 |
| 0.75 | 0 | 1.01 ±0.12 | 95.83 | 0 | 1.02 ±0.12 | 96.90 | 0 | 1.33 ±0.85 | 90.79 |
| 1.00 | 0 | 1.01 ±0.12 | 97.22 | 0 | 1.02 ±0.12 | 96.90 | 0 | 1.39 ±1.09 | 94.38 |

Table Q: **Results for InduCE-transductive on varying $\beta$:** Lower fidelity, smaller size, and higher accuracy are desired. The best results are in bold. Tr., Fid., Acc. denote transductive, fidelity and accuracy, respectively. The best result is highlighted in bold.

| $\beta$ | Tree-Cycles | | | Tree-Grid | | | BA-Shapes | | |
|---|---|---|---|---|---|---|---|---|---|
| | Fid.(%) ↓ | Size ↓ | Acc.(%) ↑ | Fid.(%) ↓ | Size ↓ | Acc.(%) ↑ | Fid.(%) ↓ | Size ↓ | Acc.(%)↑ |
| 0.0001 | 0 | 2.31 ±1.44 | 96.65 | 0 | 4.63 ±2.91 | 91.21 | 2.6 | 4.37 ±3.53 | 64.40 |
| 0.25 | 0 | 2.31 ±1.44 | 96.65 | 0 | 4.63 ±2.91 | 91.21 | 2.6 | 4.37 ±3.53 | 64.40 |
| 0.5 (default) | 0 | 2.31 ±1.44 | 96.57 | 0 | 4.67 ±2.91 | 91.05 | 2.6 | 4.37 ±3.53 | 64.40 |
| 0.75 | 0 | 2.31 ±1.44 | 96.65 | 0 | 4.63 ±2.91 | 91.21 | 2.6 | 4.37 ±3.53 | 64.40 |
| 1.00 | 0 | 2.31 ±1.44 | 96.65 | 0 | 4.63 ±2.91 | 91.21 | 2.6 | 4.37 ±3.53 | 64.40 |

Table R: **Results for InduCE-inductive on varying $\beta$:** Lower fidelity, smaller size, and higher accuracy are desired. The best results are in bold. Tr., Fid., Acc. denote transductive, fidelity and accuracy, respectively. The best result is highlighted in bold.

## K  Effect of Hyper-parameters

In this section, we examine the effect of various hyper-parameters on the performance of INDUCE on the three benchmark datasets.

**Impact of $\beta$:** In this experiment, we vary $\beta$ parameter that regulates the weightage of log-likelihood vs. the perturbation count in Equation 8. The results are provided in Tables Q and R for the transductive and inductive settings respectively. We vary $\beta$ from 0.0001 to 1. We observe that INDUCE is robust to change in $\beta$ in most cases. This can be attributed to the small sizes of counterfactuals in general in the benchmark datasets. Thus, the actual effect of $\beta$ may come into play for larger-sized counterfactuals when trajectories are longer, i.e., a larger set of edits are needed to change the class label of the target node. Further, for INDUCE-transductive we observe that for BA-Shapes datasets as $\beta$ increases the counterfactual size increases. This seems counter-intuitive at the face but hints at the fact that for smaller trajectories, it is more important to focus on reducing the log-likelihood of the label rather than increasing the weight of the counterfactual size component of the reward function. In this manner, the model is incentivized to pick perturbations that will cause the target to cross the decision boundary in the early stages of the action trajectory.

**Impact of hops ($h$):** In this experiment, we vary the number of hops in the ego-network of the target node while keeping the number of layers in the policy network constant. The results are provided in tables S and T. We vary the number of hops from 2 to 5, where 3 was the number of layers in the GNN being explained. We observe that, in general, a hop-value closer to the number of GNN layers leads to better effectiveness. This observation especially shines for INDUCE-inductive (Refer Table T) wherein increasing the hops cause a drastic drop in performance. This can be attributed to the explosion of the size of the action space or the search space of counterfactuals, which may cause generalizability issues. Furthermore, inductive explanations are harder since the explainer needs to generalize to unseen data.

**Impact of GAT-layers:** Finally, we study the impact of the number of layers in the GAT used in the policy network while keeping everything else fixed. The results are provided in Tables U and V. We observe that the number of GAT layers should be close to the number of GNN black-box layers, and thus, also the number of hops in the target's ego-network. Note that earlier we observed that increasing the number of hops in the ego network of the target node beyond the number of layers in the GNN black box reduces performance (Recall Table T). For an ego-network with a smaller number of hops, if we use a GAT with a larger number of layers, the embeddings would be over-smoothed, and that would lead to degraded performance. Therefore, ideally, for INDUCE we used 4-hop ego-network and a 3-layered GAT.

| Hops(h) | Tree-Cycles | | | Tree-Grid | | | BA-Shapes | | |
|---|---|---|---|---|---|---|---|---|---|
| | Fid.(%) ↓ | Size ↓ | Acc.(%) ↑ | Fid.(%) ↓ | Size ↓ | Acc.(%) ↑ | Fid.(%) ↓ | Size ↓ | Acc.(%)↑ |
| 2 | **0** | 1.03 ±0.16 | 99.31 | **0** | 1.14 ±0.35 | 93.80 | **0** | **1.08 0.27** | 94.74 |
| 3 | **0** | **1.01 ±0.12** | 97.22 | **0** | **1.02 ±0.12** | 96.90 | **0** | 1.12 ±0.36 | 94.96 |
| 4 (default) | **0** | **1.01 ±0.12** | 98.61 | **0** | **1.02 ±0.12** | 97.67 | **0** | 1.30 ±0.90 | **95.31** |
| 5 | **0** | **1.01 ±0.12** | **100** | **0** | **1.02 ±0.12** | **99.22** | **0** | 1.43 ±0.99 | 93.51 |

Table S: **Results for InduCE-transductive on varying hops of target's neighbourhood:** Lower fidelity, smaller size, and higher accuracy are desired. The best results are in bold. Tr., Fid., Acc. denote transductive, fidelity and accuracy, respectively. The best result is highlighted in bold.

| Hops(h) | Tree-Cycles | | | Tree-Grid | | | BA-Shapes | | |
|---|---|---|---|---|---|---|---|---|---|
| | Fid.(%) ↓ | Size ↓ | Acc.(%) ↑ | Fid.(%) ↓ | Size ↓ | Acc.(%) ↑ | Fid.(%) ↓ | Size ↓ | Acc.(%)↑ |
| 2 | **0** | **1.82 1.18** | 97.08 | 1.55 | **2.68 1.67** | **94.90** | 3.95 | **2.423 ±2.43** | **94.77** |
| 3 | **0** | 1.88 1.09 | **98.42** | **0** | 3.44 2.49 | 94.39 | 13.16 | 2.70 ±2.86 | 75.54 |
| 4 (default) | **0** | 2.31 ±1.44 | 96.57 | **0** | 4.67 ±2.91 | 91.05 | **2.6** | 4.37 ±3.53 | 64.40 |
| 5 | **0** | 2.06 ±1.39 | 96.03 | 8.52 | 5.36 ±3.82 | 91.85 | 6.58 | 3.49 ±3.02 | 83.54 |

Table T: **Results for InduCE-inductive on varying hops of target's neighbourhood:** Lower fidelity, smaller size, and higher accuracy are desired. The best results are in bold. Tr., Fid., Acc. denote transductive, fidelity and accuracy, respectively. The best result is highlighted in bold.

| Layers | Tree-Cycles | | | Tree-Grid | | | BA-Shapes | | |
|---|---|---|---|---|---|---|---|---|---|
| | Fid.(%) ↓ | Size ↓ | Acc.(%) ↑ | Fid.(%) ↓ | Size ↓ | Acc.(%) ↑ | Fid.(%) ↓ | Size ↓ | Acc.(%)↑ |
| 2 | **0** | **1.01 ±0.12** | 97.22 | **0** | **1.02 ±0.12** | 97.67 | **0** | 1.22 ±0.70 | 98.38 |
| 3 (default) | **0** | **1.01 ±0.12** | **98.61** | **0** | **1.02 ±0.12** | 97.67 | **0** | 1.30 ±0.90 | 95.31 |
| 4 | **0** | **1.01 ±0.12** | 97.22 | **0** | **1.02 ±0.12** | **99.22** | **0** | **1.18 ±0.60** | **97.32** |
| 5 | **0** | **1.01 ±0.12** | 97.22 | **0** | **1.02 ±0.12** | 97.67 | **0** | 1.32 ±0.88 | 92.543 |

Table U: **Results for InduCE-transductive on varying GAT layers in policy network:** Lower fidelity, smaller size, and higher accuracy are desired. The best results are in bold. Tr., Fid., Acc. denote transductive, fidelity and accuracy, respectively. The best result is highlighted in bold.

| Layers | Tree-Cycles | | | Tree-Grid | | | BA-Shapes | | |
|---|---|---|---|---|---|---|---|---|---|
| | Fid.(%) ↓ | Size ↓ | Acc.(%) ↑ | Fid.(%) ↓ | Size ↓ | Acc.(%) ↑ | Fid.(%) ↓ | Size ↓ | Acc.(%)↑ |
| 2 | **0** | **1.81 ±0.93** | 89.93 | 0.78 | 2.83 ±2.39 | **95.59** | 69.74 | **3.26 ±2.60** | 63.92 |
| 3 (default) | **0** | 2.31 ±1.44 | **96.57** | **0** | 4.67 ±2.91 | 91.05 | 2.6 | 4.37 ±3.53 | 64.40 |
| 4 | **0** | 2.88 ±1.48 | 66.20 | **0** | **2.80 ±1.88** | 86.76 | 3.95 | 4.27 ±4.01 | 84.85 |
| 5 | **0** | 2.81 ±1.62 | 65.45 | **0** | 4.50 ±2.35 | 76.22 | **1.31** | 3.39 ±3.58 | **95.13** |

Table V: **Results for InduCE-inductive on varying GAT layers in policy network:** Lower fidelity, smaller size, and higher accuracy are desired. The best results are in bold. Tr., Fid., Acc. denote transductive, fidelity and accuracy, respectively. The best result is highlighted in bold.

