# OpenReview forum: "InduCE: Inductive Counterfactual Explanations for Graph Neural Networks"
_TMLR — Accepted by TMLR_

### Review · Reviewer_jot5 · 2024-01-27

**Summary Of Contributions:**

The paper proposes a new counterfactual explanation method for graph neural networks. The major idea is to model the edge perturbation process as the Markov decision process and leverage the reinforcement learning framework to generate explainable structures. Experiments on several datasets demonstrate that the proposed method can outperform multiple baseline methods for the explainable node classification task. Additional analytical experiments and case studies are further provided to show the effectiveness and efficiency of the proposed method.

**Audience:**

Yes

**Broader Impact Concerns:**

No ethical concern.

**Claims And Evidence:**

No

**Requested Changes:**

See above weaknesses.

**Strengths And Weaknesses:**

Strengths
+ The problem of developing explainable graph neural networks is important and useful in real applications.

+ Proposed a new counterfactual explanation method for graph neural networks. It is reasonable to leverage the reinforcement learning framework to model the process of explainable structure generation. The proposed method achieves competitive performance in comparison with multiple baseline methods.

+ The presentation is clear and easy to follow.

Weaknesses
- Inductive Setting: The authors claimed that inductive ability is an important strength of the proposed method. However, existing parameterized algorithms can also be employed for inductive inference tasks on new nodes/graphs by utilizing the learned model, specifically the optimized parameters. I find this claim less convincing, and I believe additional demonstrations are necessary to substantiate it.

- Edge Addition: Additionally, the authors introduced the edge addition operation. While it makes sense to utilize existing sub-structures to explain prediction results, it remains unclear to me how we can interpret the model's outcomes using newly added edges, given that these edges are not originally part of the graph. Once again, I believe more demonstrations are essential to clarify this aspect.

- Task/Experiment: As far as I know, many existing methods (e.g., PGExplainer) can be applied/adapted to both node-level and graph-level tasks. Whether the proposed method be applied to graph-level tasks? If so, it would be better to see experiments/results regarding the graph classification task using existing graph-level data such as MUTAG.

---

> ### Author Response · Authors · 2024-02-22
> **Response to Reviewer jot5**
>
> **1. Inductive Setting: The authors claimed that inductive ability is an important strength of the proposed method. However, existing parameterized algorithms can also be employed for inductive inference tasks on new nodes/graphs by utilizing the learned model, specifically the optimized parameters. I find this claim less convincing, and I believe additional demonstrations are necessary to substantiate it.**
>
> *Response:* In Table 1, we summarize all the counter-factual GNN explainers in the literature. As denoted in Table 1, among existing explainers, $CF^2$ and CF_GNNExplainer are transductive and not inductive, while the other baselines of RC-explainer and CLEAR are inductive.
>
> $CF^2$ and CF-GNNExplainer cannot be used in an inductive setting for the following reason. These explainers initialize a parameter space based on the number of nodes in the input graph, resulting in an $n\times n$ weight matrix. Subsequently, this matrix is transformed into a binary $n\times n$ mask matrix, which is element-wise multiplied with the original graph's adjacency matrix to denote edge deletion (with 0s in the mask indicating deletion). As a consequence, the model's parameters are contingent upon the graph's node count, making it infeasible to apply on unseen graphs of varying sizes in an inductive setting.
>
> The discussion regarding the transductive nature of these baselines has now been integrated into Section 1 to emphasize the rationale behind their transductive behavior. It is worth highlighting that CF^2 and CF-GNNExplainer are provably transductive, making empirical demonstrations in an inductive setting unfeasible.
>
> For the inductive baselines a thorough empirical comparison is already included (please see Tables 4 and 6).
>
> **2. Edge Addition: Additionally, the authors introduced the edge addition operation. While it makes sense to utilize existing sub-structures to explain prediction results, it remains unclear to me how we can interpret the model's outcomes using newly added edges, given that these edges are not originally part of the graph. Once again, I believe more demonstrations are essential to clarify this aspect.**
>
> *Response:* Our method, InduCE provides *counter-factual explanation* for GNNs, wherein the task is to identify the *minimal change* in the graph such that the label changes. Note that this objective is different from factual explanations where the goal is to identify the substructure responsible for a prediction. Indeed, in a factual explainer, edge additions do not have a role since we need to reason within the scope of the structural components within the input graph being explained. However, in counter-factual reasoning, there is no natural reason to remain constrained to deletions only.
>
> For example, consider the molecules ethene (C2H4) and ethane (C2H6). Ethane is a saturated hydrocarbon because all carbon atoms are saturated with covalent bonds from hydrogen atoms. In contrast, in ethene, the carbon atoms are not saturated with hydrogen interactions. Saturated hydrocarbons are significantly less reactive than unsaturated hydrocarbons, leading to vastly different properties. Now, consider the counterfactual task of identifying the minimal change in an unsaturated carbon to obtain a saturated hydrocarbon. This task requires adding (saturating) C-H edges to the unsaturated carbon atoms. Thus, counterfactual explanations become more powerful. However, existing counterfactual reasoners, with the exception of CLEAR, lack the capability to add edges. We address this gap in our proposed method.
>
> To demonstrate the relevance and importance of additions in counter-factual reasoning, the updated draft now includes the following components:
>
> * The above discussion on Ethene Vs. Ethane, along with their graph representations, have now been added to Sec 1. This discussion clearly surfaces the need to incorporate additions in the search space.
> * We empirically demonstrate the need for additions to our benchmark datasets in Fig. 5b (Sec. 4.9). In this example, a node belongs to the class "Cycle" if it is part of a cycle. On the target node, which is not part of a cycle, while INDUCE is able to change its label to "Cycle", through the addition of an edge, baselines algorithms fail to do so. Considering the abundance of cyclic substructures in chemical compounds, this demonstration holds practical relevance.
> * In Figure 5a, a similar scenario is reiterated, as discussed in Section 4.9. While the baselines manage to change the label, they do so through numerous deletions instead of a single edge addition to form a cycle.
> * In Section 4.5, we statistically analyze the impact of additions on fidelity and explanation size. This experiment comprehensively establishes that with additions, we are able to explain more precisely and accurately.

---

> > ### Author Response · Authors · 2024-02-22
> > **part 2**
> >
> > **3. Task/Experiment: As far as I know, many existing methods (e.g., PGExplainer) can be applied/adapted to both node-level and graph-level tasks. Whether the proposed method be applied to **graph-level tasks**? If so, it would be better to see experiments/results regarding the graph classification task using existing graph-level data such as MUTAG.**
> >
> > *Response:* We have added graph classification in the revised manuscript (Sec. 4.10). While our focus is on explaining node classification, InduCE can be generalized to graph classification as well. To adapt InduCE pipeline for explaining graph classification predictions, we use the entire graph to construct the MDP state instead of the $\ell$-hop neighborhood of the target node. Moreover, the action space is defined by all pairs of nodes in the graph since there is no notion of a target node. The rest of the pipeline remains unchanged.
> >
> > Table 8 from our manuscript is reproduced below to present the results on graph classification explainability. We evaluate on Mutagenicity, Mutag and AIDS datasets. As visible InduCE-transductive achieves the best fidelity across all datasets, while being marginally higher in explanation size, when compared to the smallest size achieved by any of the baselines.
> >
> >
> > | Method             | Mutagenicity           |Mutagenicity| Mutag                 |Mutag| AIDS                  |AIDS|
> > |--------------------|--|------------------------|--|-----------------------|-----------------------|--|
> > |                    | **Fid.(\%)$\downarrow$**   | **Size $\downarrow$**     | **Fid.(\%)$\downarrow$**  | **Size $\downarrow$**     | **Fid.(\%)$\downarrow$**  | **Size $\downarrow$**     |
> > | **$CF^2$**         | 38.7               | 2.87 $\pm$ 1.36         | 88.2                  | **1.25 $\pm$ 0.43**     | 98.5                  | 4.83 $\pm$ 2.19         |
> > | **InduCE (trans.)**| **20.2**               | 1.30 $\pm$ 0.81     | **0**                 | 1.35 $\pm$ 0.64    | **67.8**              | 4.61 $\pm$ 2.38     |
> > | **RCExplainer** | 39.2                   | **1.02 $\pm$ 0.25**     | 47.1                  | 2.72 $\pm$ 3.4          | 91.2                  | **1.0 $\pm$ 0.0**       |
> > | **CLEAR**| OOM                    | OOM                   | 64.7                  | 16.08 $\pm$ 6.3         | 86.2              | 296.36 $\pm$ 36.57      |
> > | **InduCE (ind.)** | 45.1                   | 5.32 $\pm$ 3.85         | 20.60             | 5.0 $\pm$ 3.42          | 91.20                 | 8.91 $\pm$ 3.91         |
> >
> >
> > OOM indicates *out of memory*.
> >
> > * The auxiliary features discussed in Sec 3.1 play an important role for node classification. For graph classification, these features are not as useful and sometimes redundant (such as entropy and distribution of node labels in the neighborhood). Incorporating features relevant for graph classification is likely to improve InduCE further. We leave this as a future direction.
> >
> > * PGExplainer is a factual explainer and not counterfactual. Tables 4 and 6 demonstrate that when factual explainers are utilized for counterfactual tasks, their performance does not match that of dedicated counterfactual explainers. This finding aligns with the conclusions drawn in prior literature [1]. Superior results have been consistently reported by counterfactual explainers such as RCExplainer and $CF^2$, both of which are included as baselines in our study. Hence, PGExplainer is not considered as a baseline for graph classification.
> >
> >
> > [1] Mert Kosan, Samidha Verma, Burouj Armgaan, Khushbu Pahwa, Ambuj Singh, Sourav Medya and Sayan Ranu, GnnX-Bench: Unravelling the Utility of Perturbation-based GNN Explainers through In-depth Benchmarking, in ICLR, 2024.

---

### Review · Reviewer_YfqB · 2024-02-04

**Summary Of Contributions:**

This paper studies the counterfactual explanation problem on GNNs. It discussed some issues in existing work, including 1) lack of marginal probability modeling; 2) inability to add edges; and 3) lack of generalization.

To address these problems, this work proposes a model-agnostic, inductive counterfactual reasoning method for GNNs – InduCE, which contains two phases: training and inference. In training, a counterfactual reasoning model is learned, and in inference, the counterfactual graph is predicted based on the reasoning model.

**Audience:**

Yes

**Broader Impact Concerns:**

No concerns on the ethical implications.

**Claims And Evidence:**

Yes

**Requested Changes:**

1. Clear validation for the improvements in marginal effect modeling of perturbations
2. Need careful proofreading and improvement for presentation.

**Strengths And Weaknesses:**

Strengths:
- Point out issues of existing approaches.
- Provide theoretical analysis for the difficulty of the studied problem and the complexity of the proposed method.
- The experiments on five datasets show promising results.

Weaknesses:
- One of the major contributions of this paper is modeling the marginal effect of perturbations aid in generating better counterfactuals, which seems to be the main reason for the improvement of effectiveness. This point is important but has not been well validated. Adding theoretical analysis for the improvements of the marginal effect modeling on counterfactual explanation would be beneficial.
- Certain parts of this paper lack clarity, making them difficult to follow. Some terms, such as the sufficiency score metric, are introduced without clear explanation.
- The overall writing quality requires enhancement. Some typos are present in the current version, and informal formatting is observed throughout.

---

> ### Author Response · Authors · 2024-02-22
> **Response to Reviewer YfqB**
>
> **1. One of the major contributions of this paper is modeling the marginal effect of perturbations aid in generating better counterfactuals, which seems to be the main reason for the improvement of effectiveness. This point is important but has not been well validated. Adding theoretical analysis for the improvements of the marginal effect modeling on counterfactual explanation would be beneficial.**
>
> *Response:* Thank you for this suggestion. We have now included detailed analysis of the marginal effect in the revised manuscript. Specifically,
>
> * In Theorem 1, we establish that counterfactual reasoning is a combinatorial optimization problem by reducing it to the set cover problem. This implies that the utility of a set of edits $\mathbb{A}=\{a_1,\cdots,a_m\}$ is not simply a sum of the utilities of the individual edits , i.e., $utility(\mathbb{A})\neq\sum_{\forall a_i\in \mathbb{A}} utility(a_i)$. In our context, utility represents decrease in negative log-likelihood with respect to the original class following a set of edits.
> *  *What happens if we ignore the marginal impact of an edit, and simply follow a single-shot approach as in the baselines?* Fig. 4(b) in Sec 4.6 analyzes this question. We observe a significantly lower (lower is better) fidelity achieved by InduCE when compared to its *single-shot* counterpart and thereby establishing the importance of modeling marginal impacts. In single-shot prediction, we directly add the top-$k$ edits with the highest expected reward based on the original graph state. This contrasts with the auto-regressive pipeline of InduCE where a single edit with the highest expected reward is first added to the original graph, and then all available edits are re-prioritized based on their expected rewards on the edited graph. To ensure a fair comparison, $k$ in the single-shot approach is set to the number of edits made by InduCE until the node label changes.
>
>
> **2. Certain parts of this paper lack clarity, making them difficult to follow. Some terms, such as the sufficiency score metric, are introduced without clear explanation.**
>
> *Response:* We apologize for any confusion caused. We have conducted a comprehensive revision to improve the clarity of our writing. Regarding the terminology, we have replaced 'sufficiency' with 'fidelity'. In the context of counterfactual reasoning, sufficiency and fidelity are synonymous. The definition of fidelity can be found in Section 4.3.
>
> **3. The overall writing quality requires enhancement. Some typos are present in the current version, and informal formatting is observed throughout.**
>
> *Response:* We have fixed all typos, grammar errors, and formatting issues. If there are any still remaining, we would be happy to correct those as well. We thank the reviewer in helping us elevating the quality of our presentation.

---

### Review · Reviewer_uNdq · 2024-02-13

**Summary Of Contributions:**

The paper presents InduCE, a novel algorithm for generating counterfactual explanations in GNNs. This inductive learning framework models marginal probabilities in the perturbations space and enables both edge additions and deletions.

**Audience:**

Yes

**Claims And Evidence:**

Yes

**Requested Changes:**

Please answer and address cons and questions.

**Strengths And Weaknesses:**

Pros:
- InduCE solves the edge perturbation combinatorial optimization problem through reinforcement learning to bypass the NP-hard optimal solution computation
- The model is inductive and can support both edge addition and deletion perturbation.
- The paper is easy to follow.

Cons and questions
- In section 3.1 it says "Intuitively, the state should characterize how likely the class label of the target node v would flip following a given action". For non-binary classification problems (i.e., the number of classes is larger than 2), how is "flip" applied and can be understood? My understanding is that by "flip", the label changes. However "flip" can sound weird for multi-label scenarios.
- In section 3.2, "Gat" should be "GAT".
- The evolution of the distribution over perturbation space can be further discussed in the paper. For example, showing the width / entropy evolution across iterations.
- How does the presence and shape of the predefined motif contribute and influence the inference result?
- Can you elaborate more on the inductive nature of InduCE and how it enables the algorithm to generalize to unseen data without instance-specific training, and how it differs from the transductive approaches?

---

> ### Author Response · Authors · 2024-02-22
> **Response to Reviewer uNdq**
>
> **1. In section 3.1 it says "Intuitively, the state should characterize how likely the class label of the target node v would flip following a given action". For non-binary classification problems (i.e., the number of classes is larger than 2), how is "flip" applied and can be understood? My understanding is that by "flip", the label changes. However "flip" can sound weird for multi-label scenarios.**
>
> *Response:* This is a good suggestion. We have modified "flip" to "change" in the revised draft.
>
> **2. In section 3.2, "Gat" should be "GAT".**
>
> *Response:* Thank you for pointing this out. We have made the change.
>
> **3. The evolution of the distribution over perturbation space can be further discussed in the paper. For example, showing the width / entropy evolution across iterations.**
>
> *Response:* Evolution of perturbation space is indeed an important aspect of InduCE and we now have analyzed this aspect in detail in Section 4.6.
>
> **Single-shot vs iterative:** In single-shot prediction, we directly add the top-$k$ edits with the highest expected reward based on the original graph state. This contrasts with the auto-regressive pipeline of InduCE where a single edit with the highest expected reward is first added to the original graph, and then all available edits are re-prioritized based on their expected rewards on the edited graph. To ensure a fair comparison, $k$ in the single-shot approach is set to the number of edits made by InduCE until the node label changes. Fig 4a in Section 4.6 presents the results. We observe a significantly lower fidelity achieved by InduCE when compared to its *single-shot* counterpart and thereby establishing that the most preferred edit evolves over iterations.
>
> **Evolution of entropy:** We next analyze the evolution of the entropy of the perturbation space in Fig. 4c. We note a decrease in entropy indicating consolidation of perturbation preference till a certain number of iterations, beyond which, it starts to increase again. This trend is not entirely surprising, since a higher number of iterations indicates that InduCE has failed to change the label of the target node despite a significant number of edits, and such cases correspond to higher amount of uncertainty
>
>
> **Evolution of ranking:** How does the ranking of the edits based on their expected rewards change over iterations? Fig. 4c analyzes this question. Specifically, we compute the top-$k$ highest ranked edits at each timestep $t$, starting from $t=0$, and compute its Jaccard similarity to the top-$5$ list at $t+1$. Note that since we select the highest ranked edit in each time step, and therefore this edit cannot be present in the next timestep, the maximum jaccard similarity between the lists at two consecutive time steps is $(k-1)/(k+1)$. In Fig 4c., we note that the Jaccard similarity is consistently lower than this upper bound, indicating that the ranking list is evolving over time. Connecting back to Fig. 4a, this further highlights the need for modelling perturbations marginally instead of single-shot.
>
> **4. How does the presence and shape of the predefined motif contribute and influence the inference result?**
>
> *Response:* The complexity of the motif topology, along with the inherent difficulty of the prediction task, significantly influences the prediction accuracy of the GNN being explained and hence also affects the explainer. Specifically, when the GNN's predictions are poor due to the complexity of the task, there may be inherent uncertainty in the model functionality. This uncertainty can pose challenges for the explainer, as it must contend with ambiguity and variability in the model being explained when proposing counterfactual explanations. The explainer may struggle to provide confident and reliable explanations in the face of such uncertainty, leading to potentially inconsistent or unreliable results.
>
> Furthermore, the larger the size of the motif, the more the expected size of the explanation to make a non-motif node part of a motif. This trend is visible in Table 4, where the explanation size of Induce is largest for Tree-grid dataset, where the motif consists of 9 edges. In addition, the combinatorial space grows exponentially with respect to the size of the motif, making the explanation task harder.
>
> Finally, the shape also plays a role. The expressivity of a message-passing GNN is bounded by 1-WL tests [1]. If the GNN cannot distinguish between two shapes, this ambiguity will be passed on to the explainer as well.
>
> To fully elucidate this aspect of Induce, we have included the above discussion in Section 4.4.
>
> [1] How Powerful are Graph Neural Networks? Keyulu Xu, Weihua Hu, Jure Leskovec, Stefanie Jegelka, ICLR 2019.

---

> > ### Author Response · Authors · 2024-02-22
> > **part 2**
> >
> > **5. Can you elaborate more on the inductive nature of InduCE and how it enables the algorithm to generalize to unseen data without instance-specific training, and how it differs from the transductive approaches?**
> >
> > *Response:* We appreciate this suggestion and have incorporated them in the revised version. We detail the changes made below.
> >
> > **Difference from transductive approaches:** In Table 1, we summarize all the counter-factual GNN explainers in the literature. As denoted in Table 1, among existing explainers, $CF^2$ and CF-GNNExplainer are transductive and not inductive, while the other baselines of RC-explainer and CLEAR are inductive. First, we will explain why $CF^2$ and CF-GNNExplainer cannot be used in an inductive setting. These explainers initialize a parameter space based on the number of nodes in the input graph, resulting in an $n\times n$ weight matrix. Subsequently, this matrix is transformed into a binary $n\times n$ mask matrix, which is element-wise multiplied with the original graph's adjacency matrix to denote edge deletion (with 0s in the mask indicating deletion). As a consequence, the model's parameters are contingent upon the graph's node count, making it infeasible to apply on unseen graphs of varying sizes in an inductive setting.
> >
> > **Why InduCE is inductive?**  InduCE addresses the aforementioned limitation by ensuring that its parameter size remains independent of the graph size. Specifically, each edit involves either adding or deleting an edge. The embedding of such an edge (or action) is formed by concatenating the embeddings of its connected nodes. These node embeddings are learned through a Graph Attention Network (GAT) operating on the current graph state. Since GATs are inherently inductive—meaning their parameter set is invariant to changes in the graph size — InduCE inherits this property. This is demonstrated in Equation 10. As explained in the line following Equation 10, the parameter set dimension of InduCE, denoted by $\mathbf{W}^k\in\mathbb{R}^{d^{k-1}\times d^k}$, remains independent of the graph size. Here, $d^k$ signifies the hidden dimension in layer $k$.
> >
> > **Enhanced generalizability of InduCE:** The inductive nature of InduCE enhances its ability to make predictions on graphs of varying sizes. In essence, the embedding of an action (such as edge addition or deletion) is influenced by the local $\ell$-hop neighborhood surrounding the two nodes involved in the edit. Consequently, even when presented with an unseen graph, as long as the training data contains similar distributions of neighborhood topologies, the predictive accuracy of InduCE remains robust.
> >
> > **Changes made in revised draft to elaborate on inductive generalization:**
> > * In Section 1, we discuss in detail why some of the baseline algorithms are transductive.
> > * We have significantly elaborated Sec 3.4 to include the above discussion summarizing why InduCE is inductive, thereby facilitating generalizability across unseen graphs.

---

### Author Response · Authors · 2024-02-22
**Summary of changes in revised Manuscript**

We thank the reviewers for their insights and constructive suggestions. A comprehensive point-by-point response to the reviewers' comments is presented below. We have updated the main manuscript and the appendix to address these comments. The changes made in the main manuscript are highlighted in *blue* color. The *major additional experiments* are listed below.

* **Stronger empirical evaluation:** We have significantly enhanced the empirical validation of the proposed methodology. This includes: **(1)** evaluation of InduCE on graph classification, **(2)** better elucidating the benefit of modeling marginal impact of edits instead of the prevalent single-shot prediction approach, and **(3)** analyzing the evolution of perturbation over edit trajectories.
* **Presentation:** We have undertaken a series of presentation enhancements, which includes:
    * A more detailed explanation of why certain baselines lack inductive reasoning capabilities and highlighting the components within InduCE that address this limitation.
    * Importance and relevance of edge addition in the context of counterfactual reasoning.
    * Rectifying various typos and incorporating content suggestions provided by the reviewers.

We hope these changes address the concerns raised by the reviewers. We look forward to engaging with them further during the discussion phase and address any outstanding queries they may have.

---

> ### Author Response · Authors · 2024-02-25
> **Seeking final inputs before closure of discussion phase**
>
> Dear Reviewers,
>
> We thank you for your constructive comments on our work, which we have now incorporated in our revised manuscript. As the discussion period for our submission draws to a close in two days, we kindly inquire if there are any outstanding queries or clarifications that warrant attention. Your insights are invaluable to us, and we would greatly appreciate your input to ensure the manuscript's quality.
>
> regards,
>
> Authors

---

### Decision · Action_Editor_q5Fp · 2024-03-29

**Recommendation:** Accept with minor revision

**Comment:**

The motivation of the work is to be able to describe the minimal perturbation to a GNN graph that would enable a particular node's prediction to change. The authors tackle several aspects of this that (in combination) were not previously handled by works in this area: first, they wish for the graph perturbations to include both edge deletions *and* additions. Second, they want to build a model that need not be trained on a fixed set of nodes (and re-trained on unseen nodes). Third, they model marginal probabilities in the perturbations.

The authors propose a solution relying on an RL-based scheme. While it is fairly intricate, the authors obtain a number of high-quality results. Additionally, they have a wide variety of experiments demonstrating the value of the approach vs techniques that only include a subset of their desired properties.

Generally, the reviewers found the motivation for the paper compelling. They asked for additional clarity. The authors' updated draft is significantly clearer and is now easy to read.

Two concerns came up during the review process. One is how large of a contribution/how novel the authors' work is. Following TMLR's policy, this is not a crucial criteria (and I additionally believe there would be sufficient novelty in any case). The second concern is the theoretical value of the marginal information (vs zero-shot). The authors have done a solid job of demonstrating the impact empirically (i.e., Figure 4).

Overall, I believe the paper is worth accepting modulo some minor changes. I would like to see a revision with some minor aspects handled:
- It would be great for the authors to construct an intuitive argument or description for the value of modeling the marginal impact of edits. A full theoretical treatment (suggested by one of our reviewers) is not needed but could be a great topic for future work. Arguing for or explaining scenarios where we have reason to believe these gains are available (or conversely, will not be) is sufficient for the current work.
- While the paper has been improved clarity-wise, there are still some elements that could be further fleshed out. For example, it would be great to comment on how hyperparameter counts affect different techniques. The proposed technique in particular has a number of hyperparameters. It would be great to know how stable these are with respect to different graphs or problems, especially the reward function parameter $\beta$ and $h$, the number of hops used.

**Audience:**

Yes, counterfactuals on GNNs are a highly relevant area for the TMLR audience.

**Claims And Evidence:**

Yes. The authors have a clear theoretical setup and evaluate it through experiments that are reasonable for this setting, and through a series of useful ablations.

---

> ### Author Response · Authors · 2024-04-27
> **Summary of changes in minor revision**
>
> Dear Action Editor,
>
> Thank you for your time and efforts in coordinating the review process of our submission. We have uploaded the revised version addressing the minor concerns summarized in your decision statement. The new changes are highlighted in `teal` font color. The changes in the initial revision conducted are in `blue` font color. A summary of the changes introduced in this minor revision is provided below.
>
> * We have provided a concrete example of why modeling marginal impact is important in `Sec 1`. The example depicted in `Fig. 2` showcases how zero-shot predictions may lead to inflated counterfactual sizes. We further note that Section 4.6 shows this intuition is not hypothetical. Specifically, if InduCE is run to make zero-shot predictions, like existing algorithms, there is a significant drop in performance.
>
> * We have added experiments to study the **(1)** impact of $\beta$, **(2)** number of hops ($h$) and the **(3)** the number of GAT layers on the performance of InduCE. The detailed analysis and experiments are presented in `Appendix K` with reference from `Sec 4.2` in the main paper. The key observations that emerge are as follows:
>    * $\beta$ has minimal impact on the performance of InduCE. This can be attributed to the small sizes of counterfactuals in general in the benchmark datasets. Thus, the actual effect of $\beta$ may come into play for larger-sized counterfactuals when trajectories are longer, i.e., a larger set of edits are needed to change the class label of the target node.
>    * The number of hops and the number of layers in GAT follow a similar trend. The performance is optimized when they are similar to the number of layers used in the black box GNN being explained. If they deviate from the number of layers in the black box GNN the performance drops. This drop is more enhanced in the inductive version since the explainer needs to generalize to unseen graphs.
>
> We hope with these changes the outlined concerns are addressed. Should any further questions arise in response to our revision, we are more than willing to provide additional clarification.